# Transfer Learning in High-dimensional Ising Models

**Joonho Kim** [1]  **Seyoung Park** [2]

## Abstract

In high-dimensional Ising model estimation, target sample sizes are often limited, and effectively using auxiliary binary datasets of unknown relevance remains challenging. To address this, we propose Trans-Ising, a transfer learning method that combines a loss-based source screening rule with a two-stage estimation procedure. The method first identifies informative auxiliary sources using held-out target pseudolikelihood to prevent negative transfer. It then computes an initial estimator via pooled nodewise $\ell_1$-regularized logistic regression, followed by a target-only correction step using a folded-concave penalty. Theoretically, we establish fixed-node $\ell_2$ and $\ell_1$ error bounds, exact graph selection consistency, and the conditional consistency of the screening rule. Through extensive simulations and real-data analyses, we demonstrate that Trans-Ising achieves lower estimation errors than both target-only estimation and naive data pooling.

## 1. Introduction

The Ising model has a long history in spatial statistics and statistical physics (Besag, 1974; 1975). It is widely used for binary network data in psychometrics and related applications (van Borkulo et al., 2014; Epskamp et al., 2018; Waldorp et al., 2019; Park et al., 2022; Brusco et al., 2023), and recent work has studied statistical inference and structure learning for Ising models (Bhattacharya & Mukherjee, 2018; Lokhov et al., 2018; Meng et al., 2021). In this model, each variable represents a binary state, and the interaction network represents pairwise dependence. Estimating this edge set is a common inferential target in these applications because edges are interpreted as conditional associations among binary variables. A standard approach

for high-dimensional Ising model selection is nodewise $\ell_1$-regularized logistic regression (Ravikumar et al., 2010). This neighborhood-selection method avoids the intractable partition function and has become a common baseline in the Ising-model literature (Santhanam & Wainright, 2012; Barber & Drton, 2015; Kuang et al., 2017; De Canditiis, 2020). For example, in cancer genomics, modeling mutation profiles requires estimating interactions among $p = 200$ genes using limited target observations, such as $n_0 = 160$ samples. When $p$ is comparable to or larger than $n_0$, target-only nodewise logistic regressions estimate sparse neighborhoods from limited target observations, which increases false-positive and false-negative edge errors.

To reduce the target estimation error, transfer learning methods use related auxiliary datasets (Pan & Yang, 2010; Fawaz et al., 2018; Cai & Wei, 2021; Cai & Pu, 2024; Cai et al., 2024a;b; Weiss et al., 2016; Zhuang et al., 2021; Hosna et al., 2022). It has been thoroughly developed by Li et al. (2022) and Tian & Feng (2023) for high-dimensional linear and generalized linear models, by Kim et al. (2025) for benign-overfitting linear regression, and by Park et al. (2025) for large-scale low-rank regression, with related approaches in Li et al. (2023) and Zhao et al. (2026) for Gaussian graphical models. For Gaussian graphical models, correction-based transfer procedures reduce precision-matrix estimation error when sources share structure with the target (Zhao et al., 2026). Despite the growing availability of auxiliary binary datasets, there has been limited research on transfer learning for discrete graphical models such as the Ising model.

Extending transfer learning theory to high-dimensional Ising models presents technical difficulties. Each nodewise Ising regression uses a signed logistic pseudolikelihood, and cross-domain heterogeneity shifts the pooled population minimizer away from the target parameter. This shift alters the edge-selection behavior of standard $\ell_1$-penalized estimators. Because source relevance is unknown in practice, naive pooling increases the target validation risk and the target estimation error when incompatible sources are included.

In this article, we focus on the problem of transfer learning for high-dimensional Ising graph estimation and propose *Trans-Ising*, a two-stage procedure. The proposed construction first obtains an initial estimator by nodewise $\ell_1$-regularized logistic regression using the target data and

[1] Kim Jaechul Graduate School of Artificial Intelligence, KAIST, Seoul, Republic of Korea  [2] Department of Statistics and Data Science, Yonsei University, Seoul, Republic of Korea . Correspondence to: Seyoung Park <ishspsy@yonsei.ac.kr>.

*Proceedings of the 43rd International Conference on Machine Learning*, Seoul, South Korea. PMLR 306, 2026. Copyright 2026 by the author(s).

selected auxiliary samples, and then applies a target-only correction step using a folded-concave penalty. In addition, we develop a loss-based source screening rule using held-out target pseudolikelihood to select informative sources and avoid increasing the target validation risk.

**Contributions.** This article makes several contributions to transfer learning for Ising model estimation in a high-dimensional context:

- **Oracle Two-Step Estimation and Error Bounds:** For a known informative source set, we construct an oracle two-step nodewise estimator using pooled logistic lasso initialization and target-only correction. Let $s_j$ denote the target neighborhood size, $N$ denote the combined sample size from the target and informative auxiliary sources, and $h_j$ denote the source-to-target heterogeneity level. Theorem 1 establishes fixed-node $\ell_2$ and $\ell_1$ error bounds whose leading variance term is $\sqrt{s_j \log p / N}$ and whose additional terms depend on $h_j$.

- **Dual-Penalty Correction for Exact Support Recovery:** We analyze a dual-penalty correction step for support recovery in high-dimensional Ising models. While recent high-dimensional transfer learning methods for generalized linear models (Tian & Feng, 2023) established estimation guarantees for correction-based estimators with $\ell_1$-regularization, exact neighborhood recovery for high-dimensional Ising models requires a separate support-recovery analysis. This is because $\ell_1$ penalization introduces shrinkage on nonzero coefficients, and classical lasso selection analyses impose irrepresentable-type conditions (Zhao & Yu, 2006). Theorem 2 establishes fixed-node exact neighborhood recovery for the oracle two-step estimator under the stated beta-min, separation, empirical-curvature, local-solution, and cone conditions, without imposing an irrepresentable condition. In addition, Corollary 2 shows that the AND-symmetrized estimator recovers the target edge set when the fixed-node selection result holds uniformly over nodes.

- **Data-Driven Source Detection and Empirical Validation:** We develop a data-driven source detection algorithm constructed from out-of-sample pseudolikelihood loss. Theorem 3 establishes conditional recovery of a risk-defined population informative-source set under graph-level separation, validation-loss deviation, and adaptive-threshold calibration conditions. Finally, through simulation studies and real-data analyses involving mutation, online-transaction, and movie-rating data, we show that Trans-Ising attains a lower estimation error than target-only estimation in the reported settings, and a lower error than naive pooling in most reported settings. The simulations explicitly assess parameter estimation and graph recovery, while the real-data analyses assess held-out nodewise prediction error because the true interaction networks are unknown.

**Notation.** We define an Ising model on an undirected graph $G = (V, E)$, where $V = \{1, \ldots, p\}$ denotes the node set and $E$ denotes the edge set. We work with the $\{-1, 1\}$ coding. Let $X = (X_1, \ldots, X_p) \in \{-1, 1\}^p$ denote a random configuration with realization $x$. Let $X^{(0)}$ denote the target dataset with sample size $n_0$. We observe $S$ auxiliary datasets $\{X^{(s)}\}_{s=1}^S$, where $X^{(s)}$ has sample size $n_s$. Let $\theta^* \in \mathbb{R}^{p \times p}$ denote the true target parameter, and let $w^{(s)}$ denote the parameter of source $s$. The target and source interaction matrices are symmetric, and each has zero diagonal. The target edge set is defined as

$$E = \big\{\{j, k\} : 1 \le j < k \le p, \ \theta_{jk}^* \ne 0\big\}.$$

Let $\delta^{(s)} := \theta^* - w^{(s)}$ denote the contrast matrix. Let $\mathcal{A} \subseteq \{1, \ldots, S\}$ denote the unknown set of informative sources. We write $n_A := \sum_{s \in \mathcal{A}} n_s$ and $N := n_0 + n_A$. For node $j \in V$, let $\theta_{\setminus j}^* := (\theta_{jk}^*)_{k \ne j}$, let $S_j := \{k \ne j : \theta_{jk}^* \ne 0\}$, and let $s_j := |S_j|$. When writing logistic losses, we use the standard $0/1$ response coding $y_{j,i} := \mathbf{1}\{x_{ij}^{(0)} = 1\}$, which is equivalent to the $\{-1, 1\}$ representation and avoids sign ambiguities in the likelihood expressions. Let $\hat{\theta}$ denote the final estimate of the target interaction matrix. Let $\|\cdot\|_q$ and $\|\cdot\|_F$ denote the $\ell_q$ norm and the Frobenius norm, respectively. We write $a \lesssim b$ if $a \le Cb$ for an absolute constant $C > 0$ independent of the sample sizes and dimensions, and we write $a \asymp b$ if both $a \lesssim b$ and $b \lesssim a$ hold. We define $a \wedge b := \min(a, b)$.

**Organization.** Section 2 reviews related work and background on transfer learning and Ising model estimation. Section 3 introduces the proposed algorithms, and Section 4 presents the theoretical analysis. Section 5 presents simulations and the mutation-data study. Appendix S.2 presents additional real-data analyses, and Section 6 concludes the article.

## 2. Background

This section reviews related work and summarizes the background used in our development.

### 2.1. Related Work

Transfer learning in high-dimensional regression encompasses selective information borrowing, two-step correction, source detection, benign-overfitting interpolation, and low-rank multiple-response estimation (Li et al., 2022; Tian & Feng, 2023; Kim et al., 2025; Park et al., 2025). Lasso-based neighborhood selection, introduced for Gaussian graphical models (Meinshausen & Bühlmann, 2006), provides a standard estimator for high-dimensional Ising structure learning via nodewise $\ell_1$-regularized logistic regression (Ravikumar

et al., 2010). Building on related correction ideas for Gaussian graphical models (Li et al., 2023; Zhao et al., 2026), Tian & Feng (2023) analyzed two-step correction and source detection for high-dimensional GLMs. Other related contexts include time-dependent spatially varying Gaussian models (Greenewald et al., 2017) and heterogeneous logistic regression for grouped binary responses and not graph recovery (Kim et al., 2023). While recent meta-learning work explored $\ell_1$-regularized Ising support recovery under a random-task model (Xie & Honorio, 2024), our setting involves fixed domains with unknown relevance. We use a source-to-target bias formulation for the signed-design nodewise pseudolikelihood problem, augmenting the standard nodewise representation with loss-based source screening and a target-only folded-concave correction on updated coefficients for support recovery.

## 2.2. Transfer Learning

We briefly review the standard transfer learning method for high-dimensional linear models. Consider a target model:

$$Y^{(0)} = X^{(0)}\beta + \epsilon^{(0)}, \tag{1}$$

where $Y^{(0)} \in \mathbb{R}^{n_0}$, $X^{(0)} \in \mathbb{R}^{n_0 \times p}$, and $\beta \in \mathbb{R}^p$ is the target parameter.

We also observe $S$ auxiliary models

$$Y^{(s)} = X^{(s)}w^{(s)} + \epsilon^{(s)}, \quad s = 1, \ldots, S, \tag{2}$$

where $w^{(s)} \in \mathbb{R}^p$ is the parameter for the $s$-th source. A common structural assumption is that the contrast $\delta^{(s)} = \beta - w^{(s)}$ is "small" in a suitable sense, often in $\ell_1$ norm.

Following Tian & Feng (2023), one can define a transferring level via $\|\delta^{(s)}\|_1$ and the level-$h$ transferring set

$$\mathcal{A}_h = \{s : \|\delta^{(s)}\|_1 \le h\}, \tag{3}$$

where $h$ quantifies how close a source must be to be useful. In practice, $\mathcal{A}_h$ is unknown, and using non-informative sources can cause negative transfer. A recurring strategy in high dimensions is a two-step approach: (i) construct a stable estimator using pooled data from informative sources, and (ii) correct the resulting bias using only the target data. This logic underlies our Trans-Ising procedure.

## 2.3. Ising Models

The Ising model is a pairwise Markov Random Field that specifies dependence among binary variables through an undirected graph. Let $X = (X_1, \ldots, X_p) \in \{-1, 1\}^p$. A common specification (without external fields) is

$$P_\theta(x) = \frac{1}{Z(\theta)} \exp \left( \sum_{(i,j) \in E} \theta_{ij} x_i x_j \right), \tag{4}$$

where $\theta_{ij}$ is the interaction strength and $Z(\theta)$ is the partition function. Direct likelihood-based estimation is difficult because $Z(\theta)$ involves a sum over $2^p$ states.

The conditional distribution of a single node has a logistic form. For $j \in V$ and $x_j \in \{-1, 1\}$,

$$P(X_j = x_j \mid X_{\setminus j}) = \frac{\exp\left(2x_j \sum_{k \neq j} \theta_{jk} X_k\right)}{1 + \exp\left(2x_j \sum_{k \neq j} \theta_{jk} X_k\right)}. \tag{5}$$

Equivalently, $P(X_j = 1 \mid X_{\setminus j}) = \sigma\left(2 \sum_{k \neq j} \theta_{jk} X_k\right)$, where $\sigma(u) = (1 + e^{-u})^{-1}$. This identity motivates pseudolikelihood-based estimation and the neighborhood-selection approach.

Given data $X \in \{-1, 1\}^{n \times p}$, define the 0/1 response $y_{j,i} := \mathbf{1}\{x_{ij} = 1\}$ and the linear predictor $\eta_{j,i}(\theta_{\setminus j}) := 2 \sum_{k \neq j} \theta_{jk} x_{ik}$. Then the nodewise negative conditional log-likelihood at node $j$ can be written as

$$\ell_j(\theta_{\setminus j}; X) = \frac{1}{n} \sum_{i=1}^n \left\{ \log\left(1 + e^{\eta_{j,i}(\theta_{\setminus j})}\right) - y_{j,i}\,\eta_{j,i}(\theta_{\setminus j}) \right\}. \tag{6}$$

The standard $\ell_1$-regularized (Lasso) estimator (Tibshirani, 1996; van de Geer, 2008) solves:

$$\hat{\theta}_{\setminus j} = \operatorname*{argmin}_{\theta_{\setminus j} \in \mathbb{R}^{p-1}} \left\{ \ell_j(\theta_{\setminus j}; X) + \lambda \|\theta_{\setminus j}\|_1 \right\}. \tag{7}$$

Repeating this for $j = 1, \ldots, p$ yields an (asymmetric) collection of neighborhood estimates, which can be symmetrized to recover an undirected graph.

This pseudolikelihood/logistic representation is the starting point for our transfer learning method. Trans-Ising keeps the nodewise logistic model and uses auxiliary datasets through a selective pooling step followed by a target-only refinement.

**Remark 1** (Signed-design vs. 0/1 logistic form). *The loss $f(t) = \log(1 + e^{-t})$ used in the theory corresponds to the standard 0/1 logistic neighborhood regression after multiplying covariates by $x_{ij} \in \{-1, 1\}$ (signed-design transformation). Both forms are equivalent and yield the same nodewise conditional likelihood.*

## 3. Methodology

### 3.1. Oracle Trans-Ising Algorithm with Known Informative Sources

We first describe an oracle version of our procedure that assumes *the informative source set $\mathcal{A}$ is known*.

**Initial Estimation Using $\ell_1$-Regularized Logistic Regression.** The initial step pools the samples from informative auxiliary datasets $\{X^{(s)}\}_{s \in \mathcal{A}}$ with the primary dataset $X^{(0)}$.

Let $n_0$ be the sample size of the primary data, $n_s$ be the sample size for source dataset $s \in \mathcal{A}$, and let $n_A = \sum_{s \in \mathcal{A}} n_s$. Put $N = n_0 + n_A$, and write $n_r = n_0$ when $r = 0$ and $n_r = n_s$ when $r = s \in \mathcal{A}$.

As established in Section 2, estimation can be decomposed into $p$ neighborhood selection problems. Let $f(z) := \log(1 + e^{-z})$, and let $\lambda_w > 0$ denote the Step 1 regularization parameter. For each node $j \in V$, we estimate its neighborhood parameters $w_{\backslash j} \in \mathbb{R}^{p-1}$ by solving the pooled nodewise logistic lasso:

$$\hat{w}_{\backslash j}^A = \operatorname*{argmin}_{w_{\backslash j} \in \mathbb{R}^{p-1}} \left\{ \frac{1}{N} \sum_{r \in \{0\} \cup \mathcal{A}} \sum_{i=1}^{n_r} f\left(2 \sum_{k \neq j} w_{jk} x_{ij}^{(r)} x_{ik}^{(r)}\right) \right.$$
$$\left. + \lambda_w \sum_{k \neq j} |w_{jk}| \right\},$$
(8)

Repeating this for all $j = 1, \dots, p$ yields an initial (typically asymmetric) matrix estimate, denoted $\hat{w}^A$.

**Bias Correction Using Primary Data.** Although $\hat{w}^A$ uses auxiliary data, it can be biased when the source and target models differ. To reduce this bias, we apply a target-only correction. Let $\lambda_\delta > 0$ be the correction regularization parameter, and let $P_\lambda(\cdot)$ be the SCAD penalty with penalty level $\lambda$. For each node $j$, define

$$Q_j(\delta_{\backslash j}; \hat{w}_{\backslash j}^A) := \left\{ \frac{1}{n_0} \sum_{i=1}^{n_0} f\left(2 \sum_{k \neq j} (\hat{w}_{jk}^A + \delta_{jk}) x_{ij}^{(0)} x_{ik}^{(0)}\right) \right.$$
$$\left. + \lambda_\delta \sum_{k \neq j} |\delta_{jk}| + \sum_{k \neq j} P_\lambda(\hat{w}_{jk}^A + \delta_{jk}) \right\},$$
(9)

Let $\hat{\delta}_{\backslash j}^A$ denote the local solution of (9) returned by the optimization routine analyzed in Condition 3. $\hat{\delta}^A$ adjusts $\hat{w}^A$ toward the target distribution using only $X^{(0)}$.

**Dual Penalty for Bias Correction.** The pooled initializer $\hat{w}^A$ can suffer from shrinkage bias on strong edges. A naive application of Lasso in the correction step, as in previous GLM transfer methods (Tian & Feng, 2023), would impose double shrinkage on large coefficients and make it difficult to distinguish true signals from transfer bias. We therefore use a dual-penalty correction in (9): the $\ell_1$ penalty acts on the *correction* $\delta_{\backslash j}$ to encourage sparse adjustments, while a folded-concave SCAD penalty is applied to the *updated coefficients* $\hat{w}_{jk}^A + \delta_{jk}$. For coefficients above the SCAD threshold, the SCAD derivative is zero, which reduces additional shrinkage on strong signals and supports the selection argument in Theorem 2 (Fan & Li, 2001). See Appendix S.5.1 for the explicit piecewise form. In Section 4, Theorem 2 establishes nodewise support recovery and sign consistency for the resulting estimator under its stated small-error, beta-min, separation, empirical curvature, local-solution, and cone conditions.

---

**Algorithm 1** Oracle Trans-Ising Algorithm

1: **Input**: Primary data $X^{(0)}$ and informative auxiliary data $\{X^{(s)}\}_{s \in \mathcal{A}}$.
2: **Output**: Final coefficient estimate $\hat{\theta}$.
3: **Step 1 (Initial Estimation).** For each node $j \in V$, compute $\hat{w}_{\backslash j}^A$ by solving (8).
4: **Step 2 (Bias Correction).** For each node $j \in V$, compute $\hat{\delta}_{\backslash j}^A$ by solving (9).
5: **Step 3 (Final Estimation).** Assemble the asymmetric intermediate matrices $\hat{w}^A$ and $\hat{\delta}^A$ from the node-wise estimates; compute

$$\hat{\theta}^{\mathrm{asym}} = \hat{w}^A + \hat{\delta}^A,$$

and apply the symmetrization rule (AND rule) to $\hat{\theta}^{\mathrm{asym}}$ to obtain the final symmetric estimate $\hat{\theta}$.

---

**Optimization and Implementation.** Step 2 leads to a non-smooth, nonconvex objective because it combines an $\ell_1$ penalty on the correction with a folded-concave SCAD penalty on the updated coefficients. We optimize Step 2 using the standard *local linear approximation (LLA)* scheme for SCAD, which replaces the nonconvex penalty by a sequence of weighted $\ell_1$ surrogates.

For each node $j$, let $Q(\delta) := Q_j(\delta; \hat{w}_{\backslash j}^A)$ denote the objective in (9). Here $\ell_{0,j}$ denotes the target signed-design nodewise loss for node $j$; its explicit form is stated in Section 4. Put $\vartheta := \hat{w}_{\backslash j}^A + \delta$ and rewrite the objective as

$$\min_{\vartheta \in \mathbb{R}^{p-1}} \ell_{0,j}(\vartheta) + \lambda_\delta \|\vartheta - \hat{w}_{\backslash j}^A\|_1 + \sum_{k \neq j} P_\lambda(\vartheta_k).$$

At iteration $t$, LLA linearizes $P_\lambda$ at $\vartheta^{(t)}$ via weights $\omega_k^{(t)} := P_\lambda'(|\vartheta_k^{(t)}|)$, yielding the convex surrogate

$$\min_{\vartheta} \ell_{0,j}(\vartheta) + \sum_{k \neq j} \left(\lambda_\delta |\vartheta_k - \hat{w}_{jk}^A| + \omega_k^{(t)} |\vartheta_k|\right).$$

We solve this surrogate by proximal gradient. We stop when the coordinate KKT residual satisfies

$$\mathrm{dist}_\infty\left(0, \partial Q(\hat{\delta}_{\backslash j}^A)\right) := \inf_{\xi \in \partial Q(\hat{\delta}_{\backslash j}^A)} \|\xi\|_\infty \leq \varepsilon_{n,j},$$

as in Condition 3.

**Final Parameter Estimation.** The final interaction coefficients are obtained by combining the initial estimate and the correction. For each node pair $(j, k)$,

$$\hat{\theta}_{jk}^{\mathrm{asym}} = \hat{w}_{jk}^A + \hat{\delta}_{jk}^A.$$

This yields a complete, but still potentially asymmetric, matrix $\hat{\theta}^{\mathrm{asym}}$. A final symmetrization step is applied using

the AND rule:

$$\hat{\theta}_{jk} = \hat{\theta}_{kj} = \begin{cases} \frac{(\hat{\theta}_{jk}^{\mathrm{asym}} + \hat{\theta}_{kj}^{\mathrm{asym}})}{2} & \text{if } \hat{\theta}_{jk}^{\mathrm{asym}} \neq 0 \ \& \ \hat{\theta}_{kj}^{\mathrm{asym}} \neq 0, \\ 0, & \text{otherwise}, \end{cases}$$

which produces the final undirected graph estimate.

### 3.2. Trans-Ising Algorithm with Informative Source Detection

The oracle algorithm assumes the informative source set $\mathcal{A}$ is known; in practice it is not, and naively pooling all auxiliary datasets can cause *negative transfer*. We therefore select informative sources by comparing each source's effect on held-out target *pseudolikelihood*.

For a configuration $x \in \{-1, 1\}^p$ and a parameter matrix $\Theta$, write $\Theta_{j, \backslash j} := (\Theta_{jk})_{k \neq j}$ and define

$$q(x_j \mid x_{\backslash j}; \Theta_{j, \backslash j}) := \frac{\exp\{2 x_j \sum_{k \neq j} \Theta_{jk} x_k\}}{1 + \exp\{2 x_j \sum_{k \neq j} \Theta_{jk} x_k\}},$$

$$\ell^{\mathrm{pseudo}}(x; \Theta) := -\sum_{j=1}^{p} \log q(x_j \mid x_{\backslash j}; \Theta_{j, \backslash j}).$$

For a dataset of size $n$ and a parameter estimate $\hat{\theta}$, define the (negative) pseudolikelihood loss

$$\begin{aligned} \mathcal{L}_n^{\mathrm{pseudo}}(\hat{\theta}) &= -\frac{1}{n} \sum_{i=1}^{n} \sum_{j=1}^{p} \log q(x_{ij} \mid x_{i, \backslash j}; \hat{\theta}_{j, \backslash j}) \\ &= \frac{1}{n} \sum_{i=1}^{n} \sum_{j=1}^{p} f\left( 2 \sum_{k \neq j} \hat{\theta}_{jk} \, x_{ij} x_{ik} \right), \end{aligned} \quad (10)$$

which measures how well the implied conditional distributions fit the target data. We keep sources whose inclusion does not increase the held-out target loss beyond a tolerance threshold, and then run the oracle Trans-Ising on the selected set.

**Data Splitting and Pseudolikelihood Loss Calculation.** We perform two-fold cross-validation on the target data $X^{(0)}$, splitting it into $\{X^{(0)[1]}, X^{(0)[2]}\}$. For each fold $r \in \{1, 2\}$, let $n_{0,r} := |X^{(0)[r]}|$. We compute with nodewise logistic lasso:

$$\hat{\theta}^{(0)[-r]} : \text{ estimate on } X^{(0)} \setminus X^{(0)[r]}, \quad (11)$$

$$\hat{\theta}^{(0+s)[-r]} : \text{ estimate on } (X^{(0)} \setminus X^{(0)[r]}) \cup X^{(s)}. \quad (12)$$

Then we evaluate both estimates on the held-out validation fold $X^{(0)[r]}$:

$$\hat{\mathcal{L}}_0^{(r)} = \frac{1}{n_{0,r}} \sum_{x \in X^{(0)[r]}} \ell^{\mathrm{pseudo}}(x; \hat{\theta}^{(0)[-r]}), \quad (13)$$

$$\hat{\mathcal{L}}_s^{(r)} = \frac{1}{n_{0,r}} \sum_{x \in X^{(0)[r]}} \ell^{\mathrm{pseudo}}(x; \hat{\theta}^{(0+s)[-r]}). \quad (14)$$

---

**Algorithm 2** Trans-Ising with Source Detection

1: **Input:** Primary data $X^{(0)}$ and auxiliary datasets $\{X^{(s)}\}_{s=1}^{S}$.
2: **Output:** Final graph estimate $\hat{\theta}$.
3: Randomly split $X^{(0)}$ into 2 folds: $\{X^{(0)[1]}, X^{(0)[2]}\}$.
4: **for** $r = 1$ to 2 **do**
5:     Compute estimate in (11)
6:     Compute baseline loss in (13)
7:     **for** $s = 1$ to $S$ **do**
8:         Compute estimate in (12)
9:         Compute combined loss in (14)
10:     **end for**
11: **end for**
12: Compute average baseline loss $\bar{\mathcal{L}}_0 = \frac{1}{2} \sum_{r=1}^{2} \hat{\mathcal{L}}_0^{(r)}$.
13: Compute variance: $\hat{\sigma}^2 = \frac{1}{2-1} \sum_{r=1}^{2} (\hat{\mathcal{L}}_0^{(r)} - \bar{\mathcal{L}}_0)^2$.
14: Set threshold $\tau \leftarrow C_\tau \hat{\sigma}$        ▷ $C_\tau > 0$.
15: **for** $s = 1$ to $S$ **do**
16:     Compute scores as in (15)
17: **end for**
18: Select informative auxiliary datasets as in (16).
19: Run Algorithm 1 on $X^{(0)}$ and $\{X^{(s)}\}_{s \in \hat{\mathcal{A}}}$.

---

**Informative Source Selection.** We define the auxiliary compatibility score $\Delta_s$ as the average difference in pseudolikelihood loss across folds:

$$\Delta_s = \frac{1}{2} \sum_{r=1}^{2} \left( \hat{\mathcal{L}}_s^{(r)} - \hat{\mathcal{L}}_0^{(r)} \right). \quad (15)$$

A negative $\Delta_s$ indicates that including $X^{(s)}$ improves predictive performance on the target fold. A positive value suggests potential negative transfer.

To account for statistical fluctuations, we allow a tolerance band. Specifically, we use an adaptive threshold $\tau = C_\tau \hat{\sigma}$, where $\hat{\sigma}$ is the empirical standard deviation of $\{\hat{\mathcal{L}}_0^{(r)}\}_{r=1}^{2}$, and select

$$\hat{\mathcal{A}} = \{s : \Delta_s < \tau\}. \quad (16)$$

**Final Oracle Trans-Ising Estimation.** The final graph estimate is obtained by applying Algorithm 1 using the full target dataset $X^{(0)}$ and the selected sources $\{X^{(s)}\}_{s \in \hat{\mathcal{A}}}$. This loss-based selection is adapted from Tian & Feng (2023).

## 4. Theoretical Analysis

We establish the theoretical properties of our algorithms for a fixed node $j$. Our analysis uses high-dimensional M-estimation theory for nodewise logistic Ising selection (van de Geer, 2008; Ravikumar et al., 2010; Bühlmann & van de Geer, 2011; Negahban et al., 2012), transfer-learning arguments (Li et al., 2022; Tian & Feng, 2023; Park et al., 2025), and folded-concave penalties (Fan & Li, 2001; Zhang, 2010).

**Nodewise logistic loss (signed-design form).** For a dataset $X^{(r)} = (x_{i\ell}^{(r)})_{1 \le i \le n_r, 1 \le \ell \le p}$ with $x_{i\ell}^{(r)} \in \{-1, 1\}$, fix a node $j \in V$ and define the signed covariates $z_{i,\ell}^{(r)} := 2\, x_{ij}^{(r)} x_{i\ell}^{(r)}, \ell \in V \setminus \{j\}, z_i^{(r)} \in \mathbb{R}^{p-1}$. For $u \in \mathbb{R}^{p-1}$, define the nodewise logistic loss $\ell_j^{(r)}(u) := \frac{1}{n_r} \sum_{i=1}^{n_r} f(z_i^{(r)\top} u)$, where $f(t) := \log(1 + e^{-t})$. We write $\ell_{0,j}(\cdot) := \ell_j^{(0)}(\cdot)$ for the target loss.

**Pooled objective and population quantities.** For informative set $\mathcal{A}$, let $N := n_0 + \sum_{s \in \mathcal{A}} n_s$ and define weights $\alpha_r := n_r / N$ for $r \in \{0\} \cup \mathcal{A}$. Define the pooled empirical loss $\ell_{A,j}(u) := \sum_{r \in \{0\} \cup \mathcal{A}} \alpha_r\, \ell_j^{(r)}(u)$, and the population risks $\mathcal{L}_{r,j}(u) := \mathbb{E}[\ell_j^{(r)}(u)]$ and $\mathcal{L}_{A,j}(u) := \mathbb{E}[\ell_{A,j}(u)]$. Let $w_{A,\setminus j}^* := \arg\min_{u \in \mathbb{R}^{p-1}} \mathcal{L}_{A,j}(u)$. The pooled-to-target transfer bias is $\delta_{\setminus j}^* := \theta_{\setminus j}^* - w_{A,\setminus j}^*$. We quantify cross-domain heterogeneity by the source-to-target $\ell_1$ proximity level $h_j$ in Assumption 2. Under Assumption 3, Lemma 1 shows that $\|\delta_{\setminus j}^*\|_1 \le C h_j$ for some constant $C > 0$.

**Two-step estimator.** The oracle two-step estimator at node $j$ is defined as follows.

$$\hat{w}_{\setminus j}^A = \arg \min_{u \in \mathbb{R}^{p-1}} \left\{ \ell_{A,j}(u) + \lambda_w \|u\|_1 \right\}, \quad (17)$$

For Step 2, define

$$Q_j(\delta; \hat{w}_{\setminus j}^A) := \left\{ \ell_{0,j}(\hat{w}_{\setminus j}^A + \delta) + \lambda_\delta \|\delta\|_1 \right. \\ \left. + \sum_{k \ne j} P_\lambda(\hat{w}_{jk}^A + \delta_{jk}) \right\}, \quad (18)$$

Let $\hat{\delta}_{\setminus j}^A$ denote the local solution returned by the optimization routine and analyzed under Condition 3. We set $\hat{\theta}_{\setminus j} := \hat{w}_{\setminus j}^A + \hat{\delta}_{\setminus j}^A$. This vector is the asymmetric nodewise estimate used in the fixed-node analysis; Algorithm 1 applies the AND symmetrization after the nodewise estimates are computed.

**Matrix norms.** For a matrix $M$, define $\|M\|_1 := \max_k \sum_\ell |M_{\ell k}|$ (induced $\ell_1$ norm, max column sum), $\|M\|_\infty := \max_\ell \sum_k |M_{\ell k}|$ (induced $\ell_\infty$ norm, max row sum), $\|M\|_{\max} := \max_{\ell,k} |M_{\ell k}|$ (entrywise max norm). Then for any vector $x$, $\|Mx\|_\infty \le \|M\|_\infty \|x\|_1$ and $\|Mx\|_\infty \le \|M\|_{\max} \|x\|_1$.

**Target design representation.** Define the signed design matrix $Z_{\setminus j}^{(0)} := 2 \operatorname{diag}(x_{\cdot j}^{(0)}) X_{\setminus j}^{(0)} \in \mathbb{R}^{n_0 \times (p-1)}$, $u_{j,i}(\vartheta) := [Z_{\setminus j}^{(0)}]_{i,\cdot} \vartheta$, and let $\sigma(u) = (1 + e^{-u})^{-1}$ and $\hat{p}_{j,i}(\vartheta) := \sigma(u_{j,i}(\vartheta))$. With this convention,

$$\ell_{0,j}(\vartheta) = \frac{1}{n_0} \sum_{i=1}^{n_0} \left\{ \log\left(1 + e^{u_{j,i}(\vartheta)}\right) - u_{j,i}(\vartheta) \right\}$$

$$= \frac{1}{n_0} \sum_{i=1}^{n_0} f(u_{j,i}(\vartheta)),$$

and if we set $y_j^\dagger := \mathbf{1}_{n_0}$ (the all-ones vector), then the gradient and Hessian can be written as

$$\nabla \ell_{0,j}(\vartheta) = \frac{1}{n_0} Z_{\setminus j}^{(0)\top} \left(\hat{p}_j(\vartheta) - y_j^\dagger\right),$$

$$\nabla^2 \ell_{0,j}(\vartheta) = \frac{1}{n_0} Z_{\setminus j}^{(0)\top} W_j(\vartheta) Z_{\setminus j}^{(0)},$$

where $\hat{p}_j(\vartheta) = (\hat{p}_{j,1}(\vartheta), \ldots, \hat{p}_{j,n_0}(\vartheta))^\top$ and $W_j(\vartheta) = \operatorname{diag}(\hat{p}_{j,i}(\vartheta)(1 - \hat{p}_{j,i}(\vartheta)))$.

### 4.1. Rates and Selection Consistency of Algorithm 1

**Assumptions and Lemmas.** All technical assumptions, lemmas and their discussions used in this section are stated in Appendix S.3 and S.4. Conditions 1–4 are theorem-local inputs for empirical curvature, local optimization, and the Step 2 cone bound; they are separated from the primitive model and population assumptions.

**Theorem 1** (Transfer-type $\ell_2$ and $\ell_1$ rates for Oracle Trans-Ising). *Consider Algorithm 1 for a fixed node $j$. Suppose Assumptions 1, 2, 3, and 4 hold. Suppose Conditions 1, 2, 3, and 4 hold simultaneously with probability at least $1 - C_1 p^{-C_2}$. Choose tuning parameters*

$$\lambda_w = C_w \sqrt{\frac{\log p}{N}}, \quad \lambda_\delta = C_\delta \sqrt{\frac{\log p}{n_0}}, \quad \lambda = C_\lambda \sqrt{\frac{\log p}{n_0}},$$

*where $\lambda$ is the SCAD penalty level in (18) and $C_w, C_\delta, C_\lambda > 0$ are sufficiently large absolute constants. Assume further that*

$$\frac{\log p}{n_0} = o(1), \quad s_j \sqrt{\frac{\log p}{N}} + h_j \lesssim \lambda_\delta,$$

$$s_j \frac{\log p}{N} + \frac{\log p}{N} h_j^2 = o(1).$$

*Define*

$$r_{n,j} := C \left[ \sqrt{\frac{s_j \log p}{N}} + \left( \left[ \left(\frac{\log p}{N}\right)^{1/4} h_j^{1/2} \right] \wedge h_j \right) \right. \\ \left. + \left( \left[ \left(\frac{\log p}{n_0}\right)^{1/4} h_j^{1/2} \right] \wedge h_j \right) \right]$$

*for a sufficiently large constant $C > 0$. Then, for constants $C_0, c_0 > 0$, with probability at least $1 - C_0 p^{-c_0}$,*

$$\|\hat{\theta}_{\setminus j} - \theta_{\setminus j}^*\|_2 \le r_{n,j}, \quad \|\hat{\theta}_{\setminus j} - \theta_{\setminus j}^*\|_1 \lesssim s_j \sqrt{\frac{\log p}{N}} + h_j.$$

**Decomposition of the error bound.** Theorem 1 decomposes the final error into:

$$\underbrace{\sqrt{\frac{s_j \log p}{N}}}_{\text{pooled variance reduction}} + \underbrace{\left( \left(\frac{\log p}{N}\right)^{1/4} h_j^{1/2} \wedge h_j \right)}_{\text{Step 1 heterogeneity/approximation}}$$

$$+ \underbrace{\left( \left(\frac{\log p}{n_0}\right)^{1/4} h_j^{1/2} \wedge h_j \right)}_{\text{Step 2 correction limit}}.$$

Compared to the target-only rate $O_p(\sqrt{s_j \log p / n_0})$, the bound above is smaller when $N \gg n_0$ and the two heterogeneity terms are smaller than the variance term $\sqrt{s_j \log p / N}$. In particular, in the idealized homogeneous case $h_j = 0$, the bound reduces to the pooled rate up to constants, $\sqrt{s_j \log p / N}$. As $h_j$ grows, the heterogeneity terms dominate the variance term and a large $N$ no longer reduces the bound, consistent with the negative transfer of naive pooling under distribution shift.

Let $a > 2$ denote the SCAD shape parameter in the selection analysis, and let $\varepsilon_{n,j}$ denote the Step 2 coordinate KKT tolerance in Condition 3.

**Theorem 2** (Support recovery via SCAD regularization). *Let $\hat{\theta}_{\backslash j} = \hat{w}_{\backslash j}^A + \hat{\delta}_{\backslash j}^A$ be the estimator from (17)–(18). Define the estimated neighborhood by the* nonzero pattern

$$\hat{S}_j := \{k \neq j : \hat{\theta}_{jk} \neq 0\}.$$

*Assume the conditions of Theorem 1 hold. Let $r_{n,j}$ be defined as in Theorem 1. Suppose*

$$r_{n,j} \leq \lambda,$$

$$\lambda - \lambda_\delta - \varepsilon_{n,j} \geq C_{\text{sel}} \left( \sqrt{\frac{\log p}{n_0}} + s_j \sqrt{\frac{\log p}{N}} + h_j \right)$$

*for a sufficiently large constant $C_{\text{sel}} > 0$, and suppose*

$$|\theta_{jm}^*| \geq a\lambda + r_{n,j} \qquad \text{for all } m \in S_j.$$

*Then,*

$$\mathbb{P}\left( \begin{array}{c} \hat{S}_j = S_j, \\ \text{sign}(\hat{\theta}_{jk}) = \text{sign}(\theta_{jk}^*) \text{ for all } k \in S_j \end{array} \right) \to 1.$$

Theorem 2 establishes exact neighborhood recovery and sign consistency under a small-error regime, a separation condition between $\lambda$ and $(\lambda_\delta, \varepsilon_{n,j})$, and a beta-min condition. Compared with lasso, SCAD reduces shrinkage on large signals at the cost of nonconvex optimization and a computable stationary point (Condition 3).

**Remark 2** (**Selection Consistency without Irrepresentable Condition**). *Theorem 2 establishes support recovery without requiring the mutual incoherence (irrepresentable) condition on the Hessian matrix. The result uses the target RSC condition, beta-min condition, separation condition, Step 2 local-solution condition, and Step 2 cone condition. This follows because the SCAD penalty has a zero derivative above the threshold $a\lambda$, which removes the SCAD shrinkage term on those coordinates.*

## 4.2. Consistency of Informative Source Detection

While the Oracle Trans-Ising estimator uses a known informative source set $\mathcal{A}$, Algorithm 2 selects sources from held-out target pseudolikelihood. We state the detection result for the graph-level, risk-defined source set used by the validation rule.

For a parameter matrix $\Theta$, define the target population pseudolikelihood risk

$$\mathcal{L}_0^{\text{pseudo}}(\Theta) := \sum_{j=1}^{p} \mathcal{L}_{0,j}(\Theta_{j,\backslash j}).$$

For source $s$ and fold $r \in \{1, 2\}$, put $n_{0,-r} := n_0 - n_{0,r}$, where $n_{0,r}$ is the validation-fold size in the source-detection split, and define

$$\alpha_{0r,s} := \frac{n_{0,-r}}{n_{0,-r} + n_s}, \qquad \alpha_{sr} := \frac{n_s}{n_{0,-r} + n_s}.$$

For each $j = 1, \ldots, p$, define the fold-specific rowwise population pooled minimizer $W^{(0+s),*,r}$ through

$$w_{\backslash j}^{(0+s),*,r} := \arg \min_{u \in \mathbb{R}^{p-1}} \left\{ \alpha_{0r,s} \mathcal{L}_{0,j}(u) + \alpha_{sr} \mathcal{L}_{s,j}(u) \right\}.$$

Define

$$\mathcal{E}(s) := \frac{1}{2} \sum_{r=1}^{2} \mathcal{L}_0^{\text{pseudo}}(W^{(0+s),*,r}) - \mathcal{L}_0^{\text{pseudo}}(\theta^*).$$

Fix a separation constant $c_{\text{gap}} > 0$ and a sequence $\nu_n > 0$. The population informative set at level $(c_{\text{gap}}, \nu_n)$ is

$$\mathcal{A}_h := \{s \in \{1, \ldots, S\} : \mathcal{E}(s) \leq c_{\text{gap}} \nu_n\},$$

**Theorem 3** (Consistency of source detection). *Suppose Assumption 1 and Condition 5 hold. Let $\hat{\mathcal{A}}$ be the set selected by Algorithm 2 with threshold $\tau = C_\tau \hat{\sigma}$. Then*

$$\mathbb{P}(\hat{\mathcal{A}} = \mathcal{A}_h) \to 1.$$

**Corollary 1** (Oracle properties after source detection). *Fix a node $j$ and define $N_h := n_0 + \sum_{s \in \mathcal{A}_h} n_s$. Suppose that the assumptions and conditions of Theorem 3 hold. Suppose, in addition, that the assumptions, theorem-local conditions, tuning choices, and rate conditions of Theorem 1 hold with $\mathcal{A}$ and $N$ replaced by $\mathcal{A}_h$ and $N_h$, respectively. Let $\hat{\theta}_{\backslash j}^{\text{det}}$ be the asymmetric nodewise estimator obtained before the AND symmetrization step in the final run of Algorithm 1 inside Algorithm 2. Define, for a sufficiently large constant $C > 0$,*

$$r_{n,j}^{(h)} := C \left[ \sqrt{\frac{s_j \log p}{N_h}} + \left( \left(\frac{\log p}{N_h}\right)^{1/4} h_j^{1/2} \wedge h_j \right) \right.$$
$$\left. + \left( \left(\frac{\log p}{n_0}\right)^{1/4} h_j^{1/2} \wedge h_j \right) \right].$$

*Then, with probability tending to one,*

$$\|\hat{\theta}_{\backslash j}^{\text{det}} - \theta_{\backslash j}^*\|_2 \leq r_{n,j}^{(h)}, \qquad \|\hat{\theta}_{\backslash j}^{\text{det}} - \theta_{\backslash j}^*\|_1 \lesssim s_j \sqrt{\frac{\log p}{N_h}} + h_j.$$

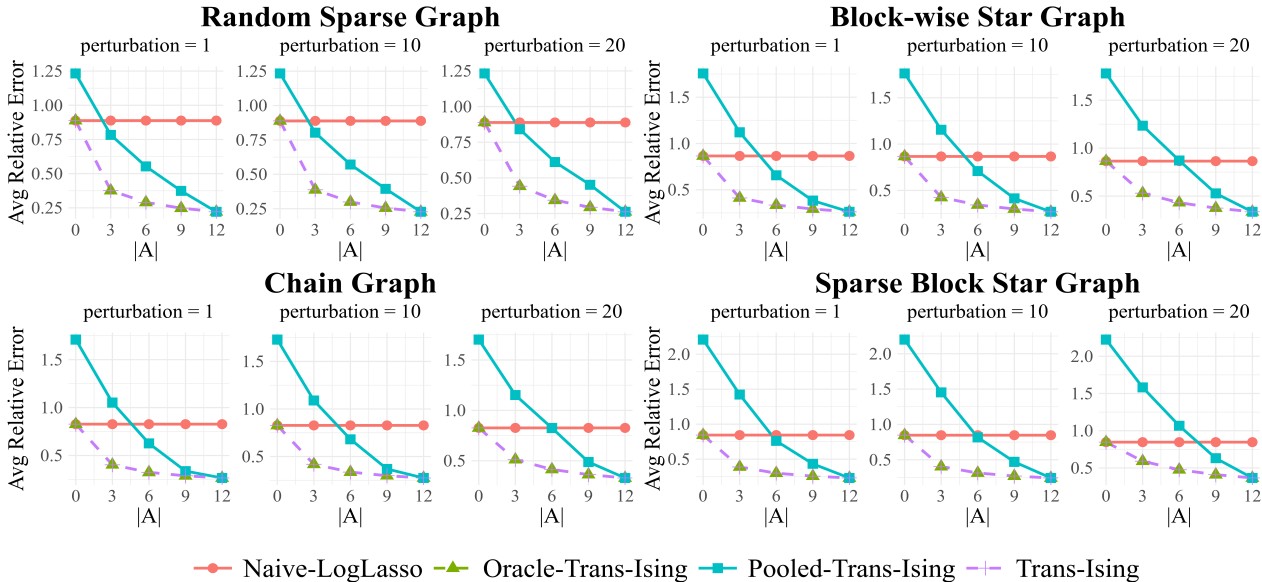

*Figure 1.* Average relative errors for Naive-LogLasso (red), Oracle Trans-Ising (green), Pooled-Trans-Ising (cyan), and Trans-Ising (purple), across four graph structures at perturbation levels $100\sigma \in \{1, 10, 20\}$, with $p = 200$.

*If the conditions of Theorem 2 also hold with $\mathcal{A}$ and $N$ replaced by $\mathcal{A}_h$ and $N_h$, respectively, and*

$$\hat{S}_j^{\text{det}} := \{k \neq j : \hat{\theta}_{jk}^{\text{det}} \neq 0\},$$

*then, with probability tending to one,*

$$\hat{S}_j^{\text{det}} = S_j, \qquad \text{sign}(\hat{\theta}_{jk}^{\text{det}}) = \text{sign}(\theta_{jk}^*) \quad \text{for all } k \in S_j.$$

*Proof.* On the event $\hat{\mathcal{A}} = \mathcal{A}_h$, Algorithm 2 runs Algorithm 1 with source set $\mathcal{A}_h$. Theorem 3 gives $\mathbb{P}(\hat{\mathcal{A}} = \mathcal{A}_h) \to 1$. Intersecting this event with the high-probability events in Theorems 1 and 2, with $\mathcal{A}$ and $N$ replaced by $\mathcal{A}_h$ and $N_h$, gives the two claims. This completes the proof. $\square$

Theorem 3 is a conditional result for the screening statistic used in Algorithm 2. It shows that graph-level separation, validation-loss deviation, and adaptive-threshold calibration imply exact recovery of $\mathcal{A}_h$. Proposition 2 gives sufficient validation-stability conditions under which Condition 5 holds.

## 5. Experiments

### 5.1. Simulation Study

We empirically evaluate Trans-Ising on synthetic Ising models and real-world applications. We assess whether Trans-Ising (i) reduces estimation error relative to target-only learning, (ii) preserves this reduction through data-driven source detection when heterogeneous sources are present, and (iii) reduces prediction error on real data. We report relative Frobenius error for parameter estimation and edge recovery performance via Precision–Recall (PR) curves (details are in Appendix S.1).

**Simulation setup.** We generate a target dataset with $n_0 = 160$ samples and $S = 12$ auxiliary datasets each with $n_s = 300$ samples in a high-dimensional regime ($p = 200 > n_0$). The target interaction matrix $\theta_{\text{true}}$ is drawn from one of four sparse graph topologies (Appendix S.1.2; Figure 4). For each run, $|\mathcal{A}| \in \{0, 3, 6, 9, 12\}$ auxiliary sources are *informative*, generated by perturbing $\theta_{\text{true}}$ with symmetric Gaussian noise at levels $\sigma \in \{0.01, 0.1, 0.2\}$. The remaining sources are generated from a structurally heterogeneous graph to induce negative transfer. All data are sampled using `IsingSampler` (Epskamp, 2025). We average results over 100 independent simulation runs.

We compare four methods: (i) *Naive-LogLasso* (target-only nodewise logistic lasso), (ii) *Pooled-Trans-Ising* (naively pooling all sources; i.e., Algorithm 1 with all auxiliaries), (iii) *Oracle Trans-Ising* (informative set known), and (iv) *Trans-Ising* (data-driven source detection). Hyperparameters are tuned proportionally to theoretical rates via cross-validation; full tuning details are in Appendix S.1.3.

**Estimation errors.** Figure 1 reports the average relative Frobenius errors across four graph structures. As the number of informative sources $|\mathcal{A}|$ increases, both Oracle Trans-Ising and Trans-Ising reduce error, consistent with variance reduction from additional compatible samples. The Trans-Ising curve closely tracks the oracle curve across the settings we considered. In contrast, pooling all sources without screening can increase error when heterogeneous

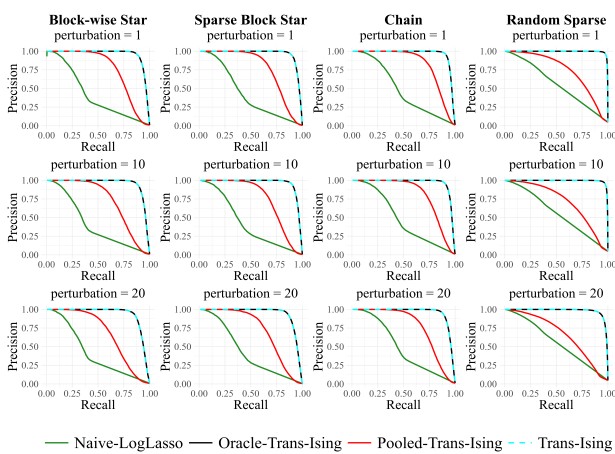

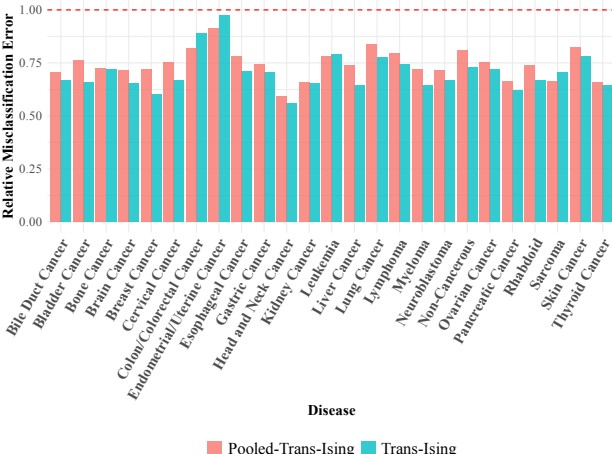

*Figure 2.* Averaged PR curves for Naive-LogLasso (green), Oracle Trans-Ising (black), Pooled-Trans-Ising (red), and Trans-Ising (cyan, dashed), across four graph structures at perturbation levels $100\sigma \in \{1, 10, 20\}$, with $p = 200$.

*Figure 3.* Relative misclassification error rates for Pooled-Trans-Ising (red) and Trans-Ising (cyan) across different target cancers. The error is relative to the Naive-LogLasso baseline (y=1.0).

(non-informative) sources are present, illustrating negative transfer in this regime.

**Edge recovery.** We evaluate support recovery using PR curves, which are typically more informative than ROC curves under graph sparsity (Appendix S.1.1). We fix $|\mathcal{A}| = 3$ informative sources and 9 heterogeneous sources, and vary the perturbation level $\sigma \in \{0.01, 0.1, 0.2\}$. In Figure 2, Trans-Ising attains PR performance comparable to the oracle benchmark across the graph families we considered. Naive pooling can yield lower precision at the recall values shown in Figure 2, consistent with additional false positives induced by heterogeneous sources. ROC curves are also reported in Appendix S.1.4 for completeness.

### 5.2. Real Data Study 1: Mutation Data Analysis

To evaluate the practical effectiveness of our proposed method, we conduct a real data study using the mutation data from the Cancer Dependency Map (DepMap) portal's `depmap_mutationCalls` dataset (release 22Q2) (Broad DepMap, 2022). Interaction-network inference is a common goal in high-dimensional biological data analysis (Marbach et al., 2012). High-dimensional binary-response modeling for Cancer Cell-Line Encyclopedia data has also been studied through low-rank multiple-response logistic regression and its joint Ising extension (Park et al., 2024). The data contain binary mutation indicators for over 18,000 genes across 1,771 cancer cell lines. We select the 200 most frequently mutated genes in the target training data. We treat each cancer type with at least 20 samples as a target study, and use the non-target groups with at least 20 samples as auxiliary studies.

Because the ground-truth interaction network is unknown,

we evaluate via prediction. For each target cancer, we run 5-fold cross-validation and report the misclassification error for predicting held-out mutation statuses. Figure 3 shows relative misclassification error compared to the Naive-LogLasso baseline (normalized to 1.0). Trans-Ising consistently improves over the target-only baseline across cancer types, while Pooled-Trans-Ising is less reliable and mostly performs worse than Trans-Ising.

To assess transferability across cancer types, Appendix S.2.2 reports the detected informative sources for each target. The number and identity of informative sources are heterogeneous, supporting the need for selective transfer and not indiscriminate pooling. Additional real data studies are provided in Appendix S.2.3–S.2.4.

## 6. Discussion

Trans-Ising provides a transfer learning method for high-dimensional Ising models that mitigates negative transfer through loss-based screening and dual-penalty correction. Theoretically, our error bounds separate pooled variance reduction from cross-domain heterogeneity ($h_j$), and achieve exact support recovery without imposing restrictive irrepresentable conditions. Empirically, Trans-Ising consistently outperforms target-only and naive-pooling baselines.

A limitation is that cross-validated screening becomes computationally costly when many auxiliaries are available. Future work may explore computationally efficient and scalable screening methods. In addition, when parameters exhibit known group structure, structured regularization such as the logistic group lasso may improve interpretability and stability (Meier et al., 2008).

## Acknowledgements

Joonho Kim and Seyoung Park's work was supported by the National Research Foundation of Korea (NRF) grant funded by the MSIT (No. RS-2025-00517793) and by the Yonsei University Research Fund of 2025-22-0071.

## Impact Statement

This paper presents work whose goal is to advance the field of Machine Learning. We do not identify specific societal consequences that require separate discussion here.

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

# APPENDIX

## S.1. Additional Experimental Details

### S.1.1. Evaluation Metrics

**Frobenius norm and relative error.** We evaluate estimation accuracy using the relative error under the Frobenius norm:

$$\frac{\|\hat{\theta} - \theta_{\text{true}}\|_F}{\|\theta_{\text{true}}\|_F},$$

where $\hat{\theta}$ is the estimated interaction matrix and $\theta_{\text{true}}$ is the ground-truth matrix.

**ROC and PR curves for graph recovery.** We evaluate edge recovery via Receiver Operating Characteristic (ROC) and Precision–Recall (PR) curves by treating edge detection as a binary classification task. For each pair $(j, k)$, we use the absolute value of the estimated parameter $|\hat{\theta}_{jk}|$ as a score, with larger scores indicating higher confidence of an edge. The ROC curve summarizes the tradeoff between

$$\text{TPR} = \frac{\text{TP}}{\text{TP} + \text{FN}} \quad \text{and} \quad \text{FPR} = \frac{\text{FP}}{\text{FP} + \text{TN}},$$

whereas the PR curve summarizes the tradeoff between precision

$$\text{Precision} = \frac{\text{TP}}{\text{TP} + \text{FP}}$$

and recall (TPR). We use AUC (ROC) and AUCPR (PR) as summary statistics.

PR curves are often more informative than ROC curves when positives are much rarer than negatives, as in sparse graph recovery (Davis & Goadrich, 2006; Saito & Rehmsmeier, 2015). In such settings, the large number of true negatives can make ROC summaries less sensitive to false positive edge discoveries, whereas PR curves focus on precision among selected edges.

### S.1.2. Simulation Design and Settings

We generate Ising model data with $p = 200$ variables. The ground-truth interaction matrix $\theta_{\text{true}}$ follows one of four sparse graph structures (Figure 4). For all structures, $\text{diag}(\theta_{\text{true}}) = 0$.

**(i) Sparse block star graph.** The $p$ nodes are divided into disjoint blocks of size 10. Within each block, one hub node is connected to a random subset of 5 spoke nodes. Edge weights are sampled from $\text{Unif}(0.5, 1.5)$.

**(ii) Block-wise star graph.** The $p$ nodes are divided into disjoint blocks of size 10. Within each block, one node is designated as a hub and connected to all other 9 nodes. Edge weights are sampled from $\text{Unif}(0.5, 1.5)$.

**(iii) Chain graph.** A path graph where only adjacent pairs $(i, i + 1)$ have nonzero interactions. Edge weights are sampled from $\text{Unif}(0.5, 1.5)$.

**(iv) Random sparse graph.** We first generate $\theta_{\text{raw}} \sim \mathcal{N}(0, 1)^{p \times p}$ and symmetrize as $(\theta_{\text{raw}} + \theta_{\text{raw}}^{\top})/4$, yielding off-diagonal entries distributed as $\mathcal{N}(0, 1/8)$. We set the diagonal to zero and induce sparsity by thresholding: off-diagonal entries with absolute value below 0.7 are set to zero.

**Target and auxiliary sampling.** We create a primary dataset with $n_0 = 160$ samples and $S = 12$ auxiliary datasets with $n_s = 300$ samples each. All Ising samples are generated using `IsingSampler` from the IsingSampler R package (Epskamp, 2025). The number of informative sources $|\mathcal{A}|$ is varied across $\{0, 3, 6, 9, 12\}$. Informative auxiliaries are generated by adding symmetric Gaussian noise $\Delta^{(s)}$ to $\theta_{\text{true}}$, where off-diagonal entries satisfy $\Delta_{jk}^{(s)} \sim \mathcal{N}(0, \sigma^2)$ and $\sigma \in \{0.01, 0.1, 0.2\}$. Non-informative auxiliaries are generated from a structurally heterogeneous graph: a sparse binary matrix is sampled with $P(\theta_{ij} = 1) = 0.05$, multiplied by standard normal noise, and symmetrized using the element-wise maximum.

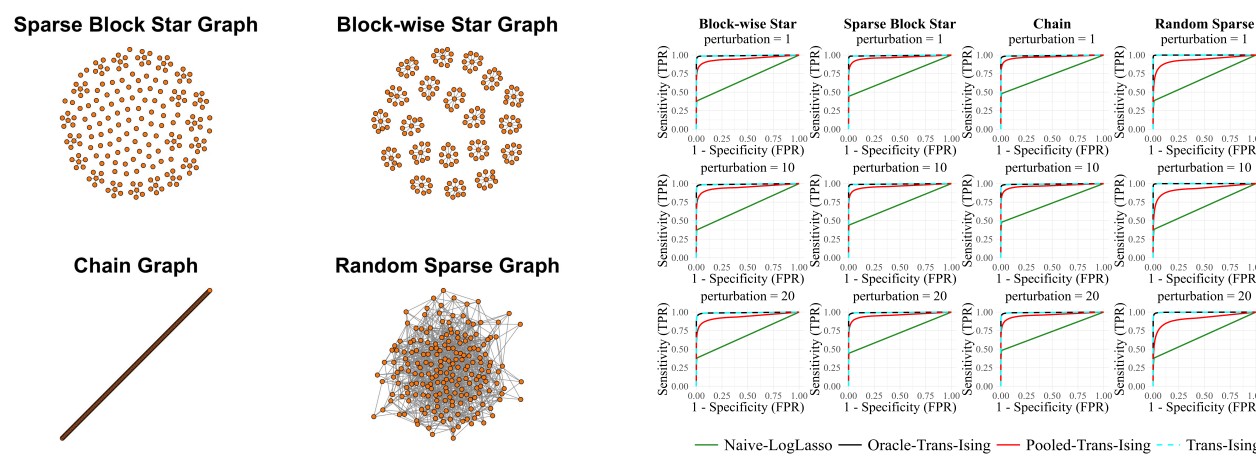

**Figure 4.** Underlying network structures for $p = 200$.

**Figure 5.** Averaged ROC curves for edge recovery across four graph structures at perturbation levels $100\sigma \in \{1, 10, 20\}$.

### S.1.3. Tuning details

Regularization parameters are tuned proportionally to theoretical rates by cross-validation. For *Oracle Trans-Ising*, put $N_{\text{orc}} := n_0 + \sum_{s \in \mathcal{A}} n_s$. For *Trans-Ising*, after source detection, put $N_{\text{sel}} := n_0 + \sum_{s \in \hat{\mathcal{A}}} n_s$ for the final estimation step. For *Pooled-Trans-Ising*, put $N_{\text{pool}} := n_0 + \sum_{s=1}^{S} n_s$.

In Step 1, we tune the penalty over the grid

$$\lambda_w \in \begin{cases} \{1.1, 0.9, 0.7, 0.5, 0.3, 0.2, 0.1\}\sqrt{\log p/N_{\text{orc}}}, & \text{Oracle Trans-Ising,} \\ \{1.1, 0.9, 0.7, 0.5, 0.3, 0.2, 0.1\}\sqrt{\log p/N_{\text{sel}}}, & \text{Trans-Ising,} \\ \{1.1, 0.9, 0.7, 0.5, 0.3, 0.2, 0.1\}\sqrt{\log p/N_{\text{pool}}}, & \text{Pooled-Trans-Ising.} \end{cases}$$

In Step 2, we first select a base penalty

$$\lambda_{\text{base}} \in \{1.5, 1.3, 1.1, 0.9, 0.7, 0.5, 0.3\}\sqrt{\frac{\log p}{n_0}}$$

by cross-validation, and then set

$$\lambda_\delta = 0.5\,\lambda_{\text{base}}, \qquad \lambda = 0.5\,\lambda_{\text{base}}.$$

The adaptive threshold parameter for source detection is set to $C_\tau = 1/2$ in $\tau = C_\tau\,\hat{\sigma}$ for all simulations. For *Naive-LogLasso*, we tune

$$\lambda_{\text{naive}} \in \{1.5, 1.4, 1.3, 1.2, 1.1, 1.0, 0.9, 0.8, 0.7, 0.6, 0.5\}\sqrt{\frac{\log p}{n_0}}$$

by cross-validation. All reported metrics are averaged over 100 independent trials for each condition.

### S.1.4. ROC curves for edge recovery

For completeness, we also provide the ROC curves (Figure 5) for the edge recovery experiments described in Section 5.1. We fix $|\mathcal{A}| = 3$ informative sources and 9 non-informative sources and vary $\sigma \in \{0.01, 0.1, 0.2\}$. The score for each potential edge is $|\hat{\theta}_{jk}|$.

## S.2. Additional Real Data Studies

### S.2.1. Exploratory Data Analysis of the Mutation Data

For better understanding of the target dataset, we perform an exploratory data analysis (EDA) to characterize the mutation landscape of the `depmap_mutationCalls` dataset.

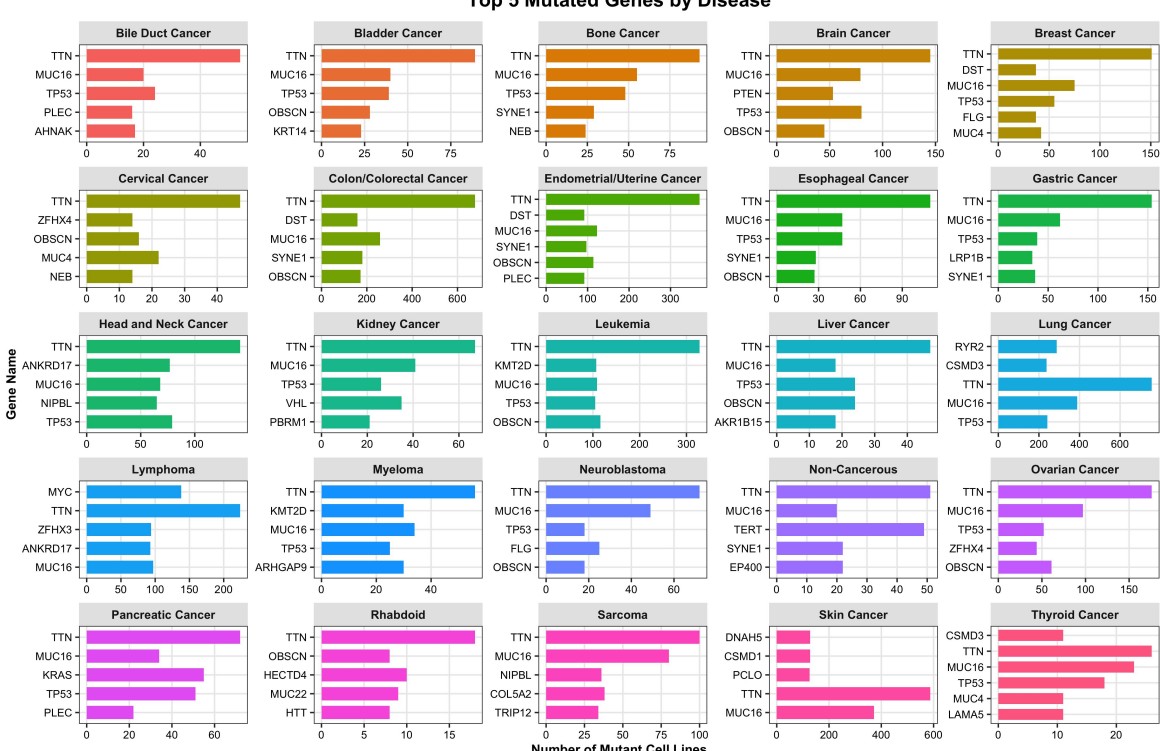

*Figure 6.* Exploratory data analysis of the DepMap mutation data. Mutation frequencies vary across primary diseases.

Figure 6 shows the most frequently mutated genes for each primary disease. Mutation frequencies vary across primary diseases. Some genes such as TTN appear frequently across cancer types, whereas the ranking and presence of other genes depend on the primary disease. For instance, KRAS is a top mutated gene in Pancreatic Cancer but not in the other reported cancer types. These patterns motivate source screening before pooling cancer-type datasets.

### S.2.2. Supplementary Analysis on Mutation Data

To further check the potential for transfer learning across different cancer types, Table 2 summarizes the outcomes of the source detection procedure for each target disease analyzed. The procedure selects sources according to the change in held-out pseudolikelihood loss. The number of informative sources varies across cancer types. For instance, Brain Cancer and Ovarian Cancer select more auxiliary datasets, whereas Skin Cancer has few informative sources. These results describe source-screening behavior and do not identify specific biological mechanisms.

*Table 1.* Disease to Index Mapping

| Disease Name | Index | Disease Name | Index | Disease Name | Index |
|---|---|---|---|---|---|
| Bile Duct Cancer | 1 | Esophageal Cancer | 9 | Myeloma | 17 |
| Bladder Cancer | 2 | Gastric Cancer | 10 | Neuroblastoma | 18 |
| Bone Cancer | 3 | Head and Neck Cancer | 11 | Non-Cancerous | 19 |
| Brain Cancer | 4 | Kidney Cancer | 12 | Ovarian Cancer | 20 |
| Breast Cancer | 5 | Leukemia | 13 | Pancreatic Cancer | 21 |
| Cervical Cancer | 6 | Liver Cancer | 14 | Rhabdoid | 22 |
| Colon/Colorectal Cancer | 7 | Lung Cancer | 15 | Sarcoma | 23 |
| Endometrial/Uterine Cancer | 8 | Lymphoma | 16 | Skin Cancer | 24 |
| | | | | Thyroid Cancer | 25 |

*Table 2.* Informative Source Indices for each Target Disease (DepMap Mutation Data).

| Target Disease | Informative Source Indices |
|---|---|
| Bile Duct Cancer | 2, 3, 4, 5, 6, 10, 11, 12, 16, 17, 18, 19, 21, 22, 23, 25 |
| Bladder Cancer | 1, 3, 4, 5, 6, 9, 10, 11, 12, 14, 15, 16, 17, 18, 19, 20, 21, 22, 23, 25 |
| Bone Cancer | 1, 2, 4, 5, 6, 9, 10, 11, 12, 14, 16, 17, 18, 19, 20, 22, 23, 25 |
| Brain Cancer | 1, 2, 3, 5, 6, 9, 10, 11, 12, 13, 14, 15, 16, 17, 18, 19, 20, 21, 22, 23, 25 |
| Breast Cancer | 2, 3, 4, 9, 11, 14, 16, 17, 18, 19, 21 |
| Cervical Cancer | 1, 4, 12, 14, 16, 17, 18, 19, 22, 23 |
| Colon/Colorectal Cancer | 1, 5, 6, 8, 9, 10, 11, 13, 14, 15, 16, 17, 18, 20, 21, 22, 23, 24, 25 |
| Endometrial/Uterine Cancer | 1, 2, 5, 6, 7, 10, 11, 13, 15, 17, 18, 20, 21, 24 |
| Esophageal Cancer | 1, 2, 4, 5, 6, 10, 11, 12, 13, 14, 15, 16, 17, 18, 19, 20, 21, 22, 23, 25 |
| Gastric Cancer | 1, 2, 3, 4, 5, 6, 9, 11, 12, 13, 14, 15, 16, 17, 18, 19, 20, 22, 23, 25 |
| Head and Neck Cancer | 1, 2, 3, 4, 5, 9, 10, 12, 15, 16, 17, 18, 19, 21, 25 |
| Kidney Cancer | 1, 2, 3, 4, 5, 6, 10, 11, 14, 16, 17, 18, 19, 20, 21, 22, 23, 25 |
| Leukemia | 1, 2, 4, 5, 9, 10, 11, 12, 14, 15, 16, 17, 18, 19, 20, 21, 22, 23, 25 |
| Liver Cancer | 1, 2, 4, 5, 6, 9, 10, 11, 12, 15, 16, 17, 18, 19, 21, 22, 25 |
| Lung Cancer | 5, 6, 14, 17, 21, 22, 25 |
| Lymphoma | 3, 4, 5, 6, 12, 14, 17, 18, 19, 22, 25 |
| Myeloma | 1, 2, 3, 4, 5, 6, 9, 10, 11, 12, 13, 14, 15, 16, 18, 19, 20, 22, 23, 25 |
| Neuroblastoma | 1, 3, 4, 5, 6, 12, 14, 16, 17, 19, 21, 22, 23, 25 |
| Non-Cancerous | 1, 2, 3, 4, 5, 6, 8, 9, 10, 11, 12, 13, 14, 16, 17, 18, 20, 22, 23, 24, 25 |
| Ovarian Cancer | 1, 2, 3, 4, 5, 6, 9, 10, 11, 12, 13, 14, 15, 16, 17, 18, 19, 21, 22, 23, 24, 25 |
| Pancreatic Cancer | 1, 2, 3, 4, 5, 6, 9, 10, 11, 12, 13, 14, 15, 16, 17, 18, 19, 20, 22, 23, 25 |
| Rhabdoid | 1, 3, 4, 5, 6, 9, 12, 14, 15, 16, 17, 18, 19, 20, 21, 23, 25 |
| Sarcoma | 1, 2, 4, 5, 9, 10, 11, 12, 15, 16, 17, 18, 19, 21, 22 |
| Skin Cancer | 9, 15, 25 |
| Thyroid Cancer | 1, 3, 4, 6, 11, 16, 18, 19, 22, 24 |

### S.2.3. Real Data Study 2: Online Transaction Analysis

We also conduct a real data study on e-commerce transaction data. We use the `UCI Online Retail` dataset from the `UCI Machine Learning Repository`, which contains transactions from a UK-based online retailer across customer countries (Chen et al., 2012; Chen, 2015).

We frame this as an Ising model estimation problem, where each item is a node, and each invoice is a sample. The binary state (1 or 0) represents whether an item was purchased in that transaction. The resulting Ising graph represents conditional associations among item-purchase indicators.

**Experimental Setup.** We define each country as a separate domain. One country with a sufficient number of invoices (e.g., Germany, France) is designated as the *target* study, and other countries with sample size greater than 50 serve as the *auxiliary* sources. We exclude the "United Kingdom" dataset because, after removing cancellation invoices and nonpositive-quantity records, it contains 18,786 of 20,728 cleaned invoices (90.6%) and would dominate the pooled analysis. These are invoice-level counts after applying the two filters and before selecting target and auxiliary countries.

For each target country, we select the $p = 200$ most frequently purchased items within that country to define the nodes of our graph. All auxiliary datasets are then projected onto this same 200-item space. We then apply our *Trans-Ising* algorithm and the *Pooled-Trans-Ising* method.

Here we use 5-fold cross-validation on the target data. We use the same nodewise conditional prediction rule as in the DepMap study, and we report the results as the relative misclassification error standardized against the *Naive-LogLasso* baseline.

**Results.** Figure 7 shows the performance of Trans-Ising (cyan bars) and Pooled-Trans-Ising (red bars) across nine target countries.

The Online Retail results are consistent with the DepMap study for the reported targets. Trans-Ising yields relative misclassification error below 1.0 for the reported targets, indicating improved prediction compared with the target-only baseline.

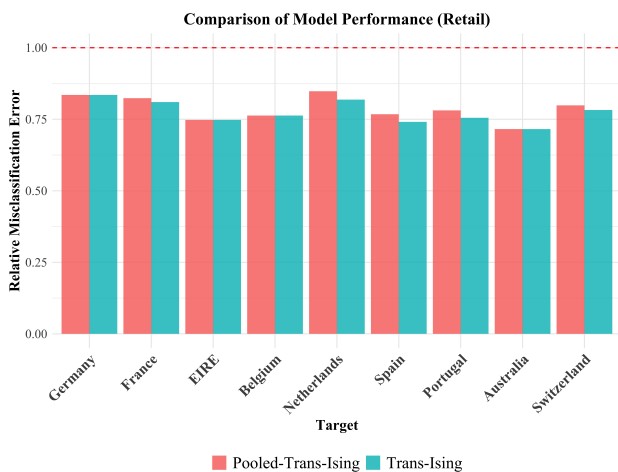
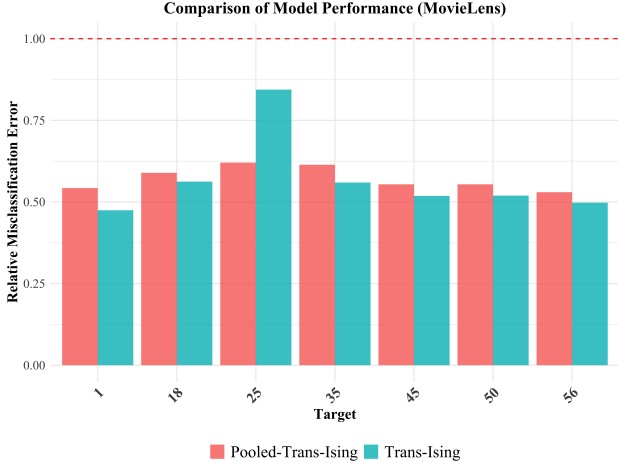

*Figure 7.* Relative misclassification error rates for Pooled-Trans-Ising and Trans-Ising on the Online Retail dataset. The error is relative to the Naive-LogLasso baseline ($y = 1.0$).

*Figure 8.* Relative misclassification error rates for Pooled-Trans-Ising and Trans-Ising on the MovieLens 1M dataset. The error is relative to the Naive-LogLasso baseline ($y = 1.0$).

*Table 3.* Informative Source Countries for each Target Country (Online Retail data).

| Target country | Informative Source Countries |
| --- | --- |
| Germany | France, EIRE, Belgium, Netherlands, Spain, Portugal, Australia, Switzerland |
| France | Germany, EIRE, Belgium, Netherlands, Spain, Portugal, Australia, Switzerland |
| EIRE | Germany, France, Belgium, Netherlands, Spain, Portugal, Australia, Switzerland |
| Belgium | Germany, France, EIRE, Netherlands, Spain, Portugal, Australia, Switzerland |
| Netherlands | Germany, France, EIRE, Spain, Portugal, Australia, Switzerland |
| Spain | Germany, France, EIRE, Belgium, Netherlands, Portugal, Australia, Switzerland |
| Portugal | Germany, France, EIRE, Netherlands, Spain, Australia, Switzerland |
| Australia | Germany, France, EIRE, Belgium, Netherlands, Spain, Portugal, Switzerland |
| Switzerland | Germany, France, EIRE, Netherlands, Spain, Portugal, Australia |

Compared with pooling all auxiliary countries, the screened estimator attains relative misclassification error comparable to or below Pooled-Trans-Ising for the reported targets, while both methods remain below the target-only baseline in Figure 7. Table 3 summarizes the selected sources for each target. In this dataset, most reported targets select many available auxiliary countries; this is an empirical observation for the included countries and not a general claim about online transaction data.

### S.2.4. Real Data Study 3: Movie Data Analysis

We also apply Trans-Ising to collaborative filtering using the `MovieLens 1M` dataset (Harper & Konstan, 2015). This dataset contains 1 million user-item ratings. We formulate this as a problem of learning user preference networks.

We define a binary *"like"* event as any rating $\geq 4$. Each user is treated as a sample, and each movie is a node. The binary state (1 or 0) indicates whether a user "liked" a specific movie. The resulting Ising graph represents conditional associations among movie-preference indicators.

**Experimental Setup.** We use user demographics to define distinct domains. For this study, we select user `Age` as the domain variable. The dataset provides discrete age brackets (1, 18, 25, 35, 45, 50, 56). We designate one age group as the *target* study and all other age groups (with $> 200$ users) as *auxiliary* sources.

Following the same methodology as in the previous studies, we select the $p = 100$ most "liked" movies based on the preferences within the target age group. All auxiliary age group datasets are then aligned to this $p = 100$ movie space. We again use the same nodewise conditional prediction rule and 5-fold cross-validation to compare *Trans-Ising* and *Pooled-Trans-Ising*, reporting the relative misclassification error against the *Naive-LogLasso* baseline.

*Table 4.* Informative source age groups for each target age group (MovieLens 1M).

| Target Age Group | Informative Source Age Groups |
|---|---|
| 1 | 35, 45, 50, 56 |
| 18 | 1 |
| 25 | 1, 45, 50, 56 |
| 35 | 1, 45, 50, 56 |
| 45 | 56 |
| 50 | 1, 45, 56 |
| 56 | 1, 45, 50 |

**Results.** Figure 8 reports 5-fold cross-validated misclassification error normalized by the target-only baseline. Trans-Ising gives a smaller relative misclassification error than naive pooling for most of the reported target age groups.

Separately, to provide a descriptive view of which age groups the screening rule deems compatible, we also report the selected source sets obtained by running the screening step on the full sample (Table 4). This full-sample analysis is not used for out-of-sample evaluation; it is included to summarize empirical patterns in estimated transferability across age groups within this dataset.

The three real-data analyses show how source screening changes prediction error for cancer genomics, online transactions, and movie ratings in the reported settings.

## S.3. Technical Assumptions for Section 4

**Assumption 1** (Independent samples across domains). *For each $r \in \{0, 1, \ldots, S\}$, the observations $X_1^{(r)}, \ldots, X_{n_r}^{(r)}$ are independent copies from the Ising distribution with parameter $\theta^{(r)}$. The samples are independent across different domains $r$. Here $\theta^{(0)} = \theta^*$, and $\theta^{(r)} = w^{(r)}$ for $r \geq 1$.*

**Assumption 2** (Sparsity and source-to-target proximity). *Fix a node $j$ and let $S_j = \{m \neq j : \theta_{jm}^* \neq 0\}$ with $s_j = |S_j|$. Assume the target neighborhood is sparse: $s_j = o\left(\frac{n_0}{\log p}\right)$. For each informative source $k \in \mathcal{A}$, let $w_{\backslash j}^{(k)}$ denote the population nodewise parameter at node $j$. Assume the informative sources are $\ell_1$-close to the target:*

$$\|\theta_{\backslash j}^* - w_{\backslash j}^{(k)}\|_1 \leq h_j, \qquad \forall k \in \mathcal{A},$$

*where $h_j \geq 0$ may depend on $j$ and is allowed to tend to zero as $(n_0, p)$ grow.*

**Assumption 3** (Information comparability across sources). *There exist a convex set $\mathcal{U} \subset \mathbb{R}^{p-1}$ and a constant $C > 0$ such that:*

1. *(**Local comparability**) for all $k \in \{0\} \cup \mathcal{A}$ and all $u, v \in \mathcal{U}$,*

$$\left\|\left(\nabla^2 \mathcal{L}_{A,j}(u)\right)^{-1} \nabla^2 \mathcal{L}_{k,j}(v)\right\|_1 \leq C.$$

   *In particular, $\nabla^2 \mathcal{L}_{A,j}(u)$ is invertible for all $u \in \mathcal{U}$.*

2. *(**Local well-posedness of population minimizers**) The population minimizers of our analysis lie in $\mathcal{U}$:*

$$\theta_{\backslash j}^* \in \mathcal{U}, \qquad w_{\backslash j}^{(k)} \in \mathcal{U} \ \forall k \in \mathcal{A}, \qquad w_{A,\backslash j}^* \in \mathcal{U},$$

   *where $w_{\backslash j}^{(k)} := \arg\min_u \mathcal{L}_{k,j}(u)$ and $w_{A,\backslash j}^* := \arg\min_u \mathcal{L}_{A,j}(u)$.*

3. *(**Integrated comparability**) The bound in (i) also holds when $\nabla^2 \mathcal{L}_{A,j}(u)$ and $\nabla^2 \mathcal{L}_{k,j}(v)$ are replaced by their averages over line segments in $\mathcal{U}$.*

**Condition 1** (Theorem-local pooled RSC and localization). *There exist constants $\kappa_A > 0$, $\tau_A \geq 0$, and a radius $r_A > 0$ such that for all*

$$\Delta \in \mathcal{C}_A := \left\{\Delta : \|\Delta_{S_j^c}\|_1 \leq 3\|\Delta_{S_j}\|_1 + 4\|(w_{A,\backslash j}^*)_{S_j^c}\|_1\right\},$$

*and all $\vartheta$ satisfying $\|\vartheta - w_{A,\backslash j}^*\|_1 \leq r_A$,*

$$\Delta^\top \nabla^2 \ell_{A,j}(\vartheta)\,\Delta \;\geq\; \kappa_A \|\Delta\|_2^2 - \tau_A \frac{\log p}{N} \|\Delta\|_1^2.$$

*In addition,*

$$\|\hat{w}_{\backslash j}^A - w_{A,\backslash j}^*\|_1 \leq r_A.$$

*This implies that the segment between $\hat{w}_{\backslash j}^A$ and $w_{A,\backslash j}^*$ is contained in $\{\vartheta : \|\vartheta - w_{A,\backslash j}^*\|_1 \leq r_A\}$.*

**Condition 2** (Theorem-local target RSC for bias correction). *Let $C_h \geq 1$ be chosen to satisfy*

$$\|\delta_{\backslash j}^*\|_1 \leq C_h h_j,$$

*which is available under Assumptions 2 and 3 by Lemma 1, and define $t_j := \left\lceil \frac{C_h\,h_j}{\lambda_\delta} \right\rceil$ . There exist constants $\kappa_0 > 0$, $\tau_0 \geq 0$, and a radius $r_0 > 0$ such that for any index set $T \subseteq V \setminus \{j\}$ with $|T| \leq t_j$, for all*

$$\Delta \in \mathcal{C}_{T,h} := \{\Delta : \|\Delta_{T^c}\|_1 \leq 3\|\Delta_T\|_1 + 3C_h h_j\},$$

*and all $\vartheta$ satisfying $\|\vartheta - \theta_{\backslash j}^*\|_1 \leq r_0$,*

$$\Delta^\top \nabla^2 \ell_{0,j}(\vartheta)\,\Delta \;\geq\; \kappa_0 \|\Delta\|_2^2 - \tau_0 \frac{\log p}{n_0} \|\Delta\|_1^2.$$

**Assumption 4** (Bounded nodewise fields and non-degenerate curvature). *The Ising variables satisfy $x_{ik} \in \{-1, 1\}$. There exists a finite constant $M < \infty$ such that, for the target parameter $\theta^*$ and all informative-source parameters $\{w^{(s)}\}_{s \in \mathcal{A}}$,*

$$\max_{r \in \{0\} \cup \mathcal{A}} \; \max_{j \in V} \; \max_{x \in \{-1,1\}^p} \; 2 \left| \sum_{k \neq j} \theta_{jk}^{(r)} x_k \right| \;\leq\; M,$$

*where $\theta^{(0)} := \theta^*$ and $\theta^{(r)} := w^{(r)}$ for $r \in \mathcal{A}$. For any signed linear predictor whose absolute value is bounded by $M$, define*

$$\rho_0 \;:=\; \inf_{|t| \leq M} f''(t) \;>\; 0, \qquad f(t) = \log(1 + e^{-t}).$$

*Curvature over the empirical neighborhoods used in the proofs is imposed separately in Conditions 1 and 2.*

**Condition 3** (Theorem-local stationary local solution in Step 2). *Let $Q(\delta) := Q_j(\delta; \hat{w}_{\backslash j}^A)$ denote the Step 2 objective in (18). Assume the optimization routine returns a point $\hat{\delta}_{\backslash j}^A$ and a tolerance $\varepsilon_{n,j} \geq 0$ such that:*

1. *(**Approximate stationarity**)*
$$\mathrm{dist}_\infty\big(0, \partial Q(\hat{\delta}_{\backslash j}^A)\big) := \inf_{\xi \in \partial Q(\hat{\delta}_{\backslash j}^A)} \|\xi\|_\infty \leq \varepsilon_{n,j}.$$

   *Equivalently, for each coordinate $k \neq j$, there exist $g_{jk} \in \partial|\hat{\delta}_{jk}^A|$ and $h_{jk} \in \partial P_\lambda(\hat{w}_{jk}^A + \hat{\delta}_{jk}^A)$ with $|g_{jk}| \leq 1$ such that*
$$\left| \left[ \nabla \ell_{0,j}(\hat{w}_{\backslash j}^A + \hat{\delta}_{\backslash j}^A) \right]_k + \lambda_\delta\, g_{jk} + h_{jk} \right| \leq \varepsilon_{n,j}.$$

2. *(**Local minimality in a neighborhood of $\delta^*$**) There exists a radius $r_\delta > 0$ such that $\|\hat{\delta}_{\backslash j}^A - \delta_{\backslash j}^*\|_1 \leq r_\delta$ and*
$$Q(\hat{\delta}_{\backslash j}^A) \leq Q(\delta) \qquad \text{for all } \delta \text{ satisfying } \|\delta - \delta_{\backslash j}^*\|_1 \leq r_\delta.$$

3. *(**RSC neighborhood applicability**) Both the oracle-shifted point and the returned estimator lie in the target-RSC neighborhood in Condition 2, i.e.,*
$$\left\| \hat{w}_{\backslash j}^A + \delta_{\backslash j}^* - \theta_{\backslash j}^* \right\|_1 \leq r_0, \qquad \left\| \hat{w}_{\backslash j}^A + \hat{\delta}_{\backslash j}^A - \theta_{\backslash j}^* \right\|_1 \leq r_0,$$

   *where $r_0$ is the radius in Condition 2.*

**Condition 4** (Theorem-local Step 2 cone condition)**.** *Let $v = \hat{\delta}_{\backslash j}^A - \delta_{\backslash j}^*$ and $T = \{k \neq j : |\delta_{jk}^*| > \lambda_\delta\}$. The Step 2 error satisfies*

$$\|v_{T^c}\|_1 \leq 3\|v_T\|_1 + 3C_h h_j.$$

**Assumption 5** (Tuning and Beta-min for Oracle Property)**.** *Let $P_\lambda(\cdot)$ be the SCAD penalty with parameter $a > 2$. Choose tuning parameters as*

$$\lambda_w = c_w \sqrt{\frac{\log p}{N}}, \qquad \lambda_\delta = c_\delta \sqrt{\frac{\log p}{n_0}}, \qquad \lambda = c_\lambda \sqrt{\frac{\log p}{n_0}},$$

*where $c_w, c_\delta, c_\lambda$ are sufficiently large. Define the target estimation error rate $r_{n,j}$ as in Theorem 1.*

*We assume the beta-min condition:*

$$|\theta_{jm}^*| \geq a\lambda + r_{n,j} \qquad \text{for all } m \in S_j.$$

*Remark: This condition ensures that the true signals are large enough to fall into the constant penalty region of SCAD (where $P_\lambda'(\cdot) = 0$) even after accounting for estimation error, which avoids the irrepresentable condition (Zhao & Yu, 2006; Wainwright, 2009).*

**Condition 5** (Theorem-local source-detection separation and threshold calibration)**.** *For the constant $c_{\mathrm{gap}} > 0$, the sequence $\nu_n > 0$, and the set $\mathcal{A}_h$ defined before Theorem 3, there exist a constant $C > 0$ and a sequence $\epsilon_n = o(\nu_n)$ such that:*

1. *(**Informative sources**) For every $s \in \mathcal{A}_h$,*

$$\mathcal{E}(s) \leq c_{\mathrm{gap}} \nu_n.$$

2. *(**Non-informative sources**) For every $s \notin \mathcal{A}_h$,*

$$\mathcal{E}(s) \geq (1 + c_{\mathrm{gap}}) \nu_n.$$

3. *(**Validation-loss deviation**)*

$$\max_{1 \leq s \leq S} |\Delta_s - \mathcal{E}(s)| \leq C\epsilon_n$$

   *with probability tending to one.*

4. *(**Adaptive-threshold calibration**) The threshold $\tau = C_\tau \hat{\sigma}$ in Algorithm 2 satisfies*

$$c_{\mathrm{gap}} \nu_n + C\epsilon_n < C_\tau \hat{\sigma} < (1 + c_{\mathrm{gap}}) \nu_n - C\epsilon_n$$

   *with probability tending to one.*

Conditions 1–4 and Condition 5 are theorem-local high-level inputs. They are not asserted to follow from Assumptions 1–4 alone. The following propositions give sufficient conditions for the empirical curvature and validation-loss deviation inputs. The Step 2 local-solution and cone inputs remain theorem-local algorithmic conditions unless a separate optimization argument verifies Conditions 3 and 4.

### S.3.1. Sufficient Conditions for Theorem-Local Inputs

For a fixed node $j$ and each domain $r \in \{0\} \cup \mathcal{A}$, let $Z_{\backslash j}^{(r)} \in \mathbb{R}^{n_r \times (p-1)}$ denote the signed design matrix whose $i$-th row is $(z_i^{(r)})^\top$. Recall that $\ell_{A,j}$ and $\ell_{0,j}$ are the pooled and target losses. Define

$$\widehat{\Sigma}_A := \sum_{r \in \{0\} \cup \mathcal{A}} \alpha_r \frac{(Z_{\backslash j}^{(r)})^\top Z_{\backslash j}^{(r)}}{n_r}, \qquad \widehat{\Sigma}_0 := \frac{(Z_{\backslash j}^{(0)})^\top Z_{\backslash j}^{(0)}}{n_0},$$

and let $\Sigma_A := \mathbb{E}\widehat{\Sigma}_A$ and $\Sigma_0 := \mathbb{E}\widehat{\Sigma}_0$. For radii $r_A, r_0, r_\delta > 0$, define the local sets

$$\mathcal{U}_A(r_A) := \{\vartheta : \|\vartheta - w_{A,\backslash j}^*\|_1 \leq r_A\}, \qquad \mathcal{U}_0(r_0) := \{\vartheta : \|\vartheta - \theta_{\backslash j}^*\|_1 \leq r_0\}.$$

Let $\mathcal{G}_j$ denote the event on which the two empirical Gram inequalities in (19) below hold.

**Assumption 6** (Primitive bounded-design curvature inputs). *There exist constants $M_A, M_0 < \infty$ and $\gamma_A, \gamma_0 > 0$ such that:*

1. **(Local bounded fields)** *For every $\vartheta \in \mathcal{U}_A(r_A)$ and every $x \in \{-1, 1\}^p$,*

$$\left| 2x_j \sum_{k \neq j} \vartheta_k x_k \right| \leq M_A.$$

   *For every $\vartheta \in \mathcal{U}_0(r_0)$ and every $x \in \{-1, 1\}^p$,*

$$\left| 2x_j \sum_{k \neq j} \vartheta_k x_k \right| \leq M_0.$$

2. **(Population restricted eigenvalues)** *For every $\Delta \in \mathcal{C}_A$,*

$$\Delta^\top \Sigma_A \Delta \geq \gamma_A \|\Delta\|_2^2.$$

   *For every index set $T \subseteq V \setminus \{j\}$ with $|T| \leq t_j$ and every $\Delta \in \mathcal{C}_{T,h}$,*

$$\Delta^\top \Sigma_0 \Delta \geq \gamma_0 \|\Delta\|_2^2.$$

3. **(Sparse signed-design tails)** *Put*

$$m_A := \left\lfloor c_m \frac{N}{\log p} \right\rfloor, \qquad m_0 := \left\lfloor c_m \frac{n_0}{\log p} \right\rfloor$$

   *for an absolute constant $c_m > 0$ chosen such that the support union bound in the proof of Proposition 1 is bounded by $Cp^{-c}$. There exists a constant $K < \infty$ such that, for every $r \in \{0\} \cup \mathcal{A}$ and every vector $v$ with $\|v\|_2 = 1$ and $\|v\|_0 \leq 2(m_A \vee m_0)$,*

$$\|v^\top z_i^{(r)}\|_{\psi_2} \leq K,$$

   *where $\|\cdot\|_{\psi_2}$ denotes the sub-Gaussian Orlicz norm.*

4. **(Sample-size scaling)**

$$m_A \to \infty, \qquad m_0 \to \infty, \qquad \frac{\log p}{N} \to 0, \qquad \frac{\log p}{n_0} \to 0.$$

**Condition 6** (Sufficient local estimator and LLA inputs).    *1. **(Local estimator containment)** On $\mathcal{G}_j$, the Step 1 solution satisfies*

$$\|\hat{w}_{\backslash j}^A - w_{A,\backslash j}^*\|_1 \leq r_A.$$

2. **(LLA local-basin condition)** *Let $T = \{k \neq j : |\delta_{jk}^*| > \lambda_\delta\}$ and*

$$\mathcal{B}_{\delta,j} := \left\{ \delta : \|\delta - \delta_{\backslash j}^*\|_1 \leq r_\delta, \ \|(\delta - \delta_{\backslash j}^*)_{T^c}\|_1 \leq 3\|(\delta - \delta_{\backslash j}^*)_T\|_1 + 3C_h h_j \right\}.$$

*The Step 2 LLA routine is initialized in $\mathcal{B}_{\delta,j}$, all surrogate problems are solved to coordinate KKT tolerance $\varepsilon_{n,j}$, the iterates remain in $\mathcal{B}_{\delta,j}$, and the returned point $\hat{\delta}_{\backslash j}^A$ satisfies the original Step 2 KKT residual bound*

$$\mathrm{dist}_\infty\big(0, \partial Q(\hat{\delta}_{\backslash j}^A)\big) \leq \varepsilon_{n,j}.$$

*In addition, the returned point satisfies*

$$Q(\hat{\delta}_{\backslash j}^A) \leq Q(\delta) \qquad \textit{for all } \delta \textit{ such that } \|\delta - \delta_{\backslash j}^*\|_1 \leq r_\delta.$$

*In addition,*

$$\left\|\hat{w}_{\backslash j}^A + \delta_{\backslash j}^* - \theta_{\backslash j}^*\right\|_1 \leq r_0, \qquad \left\|\hat{w}_{\backslash j}^A + \hat{\delta}_{\backslash j}^A - \theta_{\backslash j}^*\right\|_1 \leq r_0.$$

**Proposition 1** (Verification of empirical curvature inputs). *Suppose Assumptions 1, 2, 3, 4, and 6 and Condition 6(i) hold. Then Conditions 1 and 2 hold with probability at least $1 - Cp^{-c}$ for some constants $C, c > 0$. On this event, Condition 6(ii) implies Conditions 3 and 4.*

*Proof.* We first verify the empirical Gram bounds. For $q = A$, set $\widehat{\Sigma}_q = \widehat{\Sigma}_A$, $\Sigma_q = \Sigma_A$, $n_q = N$, $m_q = m_A$, and $\gamma_q = \gamma_A$. For $q = 0$, set $\widehat{\Sigma}_q = \widehat{\Sigma}_0$, $\Sigma_q = \Sigma_0$, $n_q = n_0$, $m_q = m_0$, and $\gamma_q = \gamma_0$. For a fixed $q \in \{A, 0\}$ and a fixed support $B$ with $|B| \leq 2m_q$, let $\mathbb{S}_B$ be the unit sphere on $B$ and let $\mathcal{N}_B$ be a $1/4$-net of $\mathbb{S}_B$ with $|\mathcal{N}_B| \leq 9^{|B|}$. For any $u \in \mathcal{N}_B$, the variables $(u^\top z_i^{(r)})^2 - \mathbb{E}\{(u^\top z_i^{(r)})^2\}$ are independent and sub-exponential with norm bounded by a constant that depends only on $K$. For $q = A$ these variables are summed over $r \in \{0\} \cup \mathcal{A}$ with weight $1/N$, and for $q = 0$ they are summed over the target sample with weight $1/n_0$. Bernstein's inequality gives

$$\Pr\left(\left|u^\top(\widehat{\Sigma}_q - \Sigma_q)u\right| > \gamma_q/16\right) \leq 2\exp(-cn_q)$$

for a constant $c > 0$ that depends only on $K$ and $\gamma_q$. A union bound over the nets and over all supports with $|B| \leq 2m_q$, together with $m_q \leq c_m n_q / \log p$ and a sufficiently small $c_m$, gives, with probability at least $1 - Cp^{-c}$,

$$\sup_{\|u\|_2 = 1, \ \|u\|_0 \leq 2m_q} \left|u^\top(\widehat{\Sigma}_q - \Sigma_q)u\right| \leq \gamma_q/4$$

simultaneously for $q = A$ and $q = 0$. To pass from sparse vectors to all vectors, fix $\Delta$ and order its coordinates by decreasing absolute value. Let $S_0, S_1, \ldots$ be consecutive blocks of size $m_q$. The sparse bound above and polarization imply

$$\left|a^\top(\widehat{\Sigma}_q - \Sigma_q)b\right| \leq C\gamma_q\|a\|_2\|b\|_2$$

whenever $|\operatorname{supp}(a) \cup \operatorname{supp}(b)| \leq 2m_q$. Because

$$\sum_{\ell \geq 1} \|\Delta_{S_\ell}\|_2 \leq m_q^{-1/2}\|\Delta\|_1,$$

expanding $\Delta^\top(\widehat{\Sigma}_q - \Sigma_q)\Delta$ over the blocks gives

$$\left|\Delta^\top(\widehat{\Sigma}_q - \Sigma_q)\Delta\right| \leq \frac{\gamma_q}{2}\|\Delta\|_2^2 + C\frac{\log p}{n_q}\|\Delta\|_1^2.$$

Combining this deviation bound with Assumption 6(ii) gives constants $C, c > 0$ such that, with probability at least $1 - Cp^{-c}$,

$$\begin{aligned}
\Delta^\top\widehat{\Sigma}_A\Delta &\geq \frac{\gamma_A}{2}\|\Delta\|_2^2 - C\frac{\log p}{N}\|\Delta\|_1^2 \qquad \text{for all } \Delta \in \mathcal{C}_A, \\
\Delta^\top\widehat{\Sigma}_0\Delta &\geq \frac{\gamma_0}{2}\|\Delta\|_2^2 - C\frac{\log p}{n_0}\|\Delta\|_1^2
\end{aligned} \tag{19}$$

for all $\Delta \in \mathcal{C}_{T,h}$ and all $T \subseteq V \setminus \{j\}$ satisfying $|T| \leq t_j$.

For every $\vartheta \in \mathcal{U}_A(r_A)$, Assumption 6(i) gives

$$f''\{(z_i^{(r)})^\top\vartheta\} \geq \rho_A := \inf_{|u| \leq M_A} f''(u) > 0.$$

This and (19) imply

$$\Delta^\top\nabla^2\ell_{A,j}(\vartheta)\Delta \geq \rho_A\Delta^\top\widehat{\Sigma}_A\Delta \geq \frac{\rho_A\gamma_A}{2}\|\Delta\|_2^2 - C\rho_A\frac{\log p}{N}\|\Delta\|_1^2$$

for all $\Delta \in \mathcal{C}_A$ and all $\vartheta \in \mathcal{U}_A(r_A)$. Together with Condition 6(i), this verifies Condition 1 with $\kappa_A = \rho_A\gamma_A/2$ and $\tau_A = C\rho_A$. The same argument with $\rho_{0,\mathrm{loc}} := \inf_{|u| \leq M_0} f''(u) > 0$ verifies Condition 2 with $\kappa_0 = \rho_{0,\mathrm{loc}}\gamma_0/2$ and $\tau_0 = C\rho_{0,\mathrm{loc}}$.

It remains to relate Condition 6(ii) to the Step 2 theorem-local inputs. Condition 6(ii) gives the original-objective coordinate KKT tolerance, the local minimality condition, and the target-RSC neighborhood condition in Condition 3. Because the returned point belongs to $\mathcal{B}_{\delta,j}$, the vector $v = \hat{\delta}_{\backslash j}^A - \delta_{\backslash j}^*$ satisfies the cone bound in Condition 4. This completes the proof. $\square$

For source detection, let $\widetilde{W}^{(0),r}$ denote the target-only training-fold estimator used in fold $r$ of Algorithm 2, and let $\widetilde{W}^{(0+s),r}$ denote the corresponding target-plus-source estimator for source $s$. Put $n_{\min}^{\mathrm{val}} := \min_{r=1,2} n_{0,r}$.

**Condition 7** (Sufficient validation inputs). *There exist sequences $b_n \geq 0$, $\sigma_n > 0$, and $\epsilon_n$ such that*

$$\epsilon_n = b_n + p\sqrt{\frac{\log(S \vee p)}{n_{\min}^{\mathrm{val}}}}, \qquad \epsilon_n = o(\nu_n).$$

*With probability at least $1 - Cp^{-c}$, the following conditions hold:*

1. ***(Uniform bounded fitted fields)*** *For every node $j$, every fold $r$, and every fitted matrix*

$$\Theta \in \{\widetilde{W}^{(0),r}\} \cup \{\widetilde{W}^{(0+s),r} : 1 \leq s \leq S\},$$

   *the nodewise signed linear predictor satisfies*

$$\sup_{x \in \{-1,1\}^p} \left| 2x_j \sum_{k \neq j} \Theta_{jk} x_k \right| \leq B_v$$

   *for a finite constant $B_v$.*

2. ***(Training-estimator risk stability)***

$$\left| \frac{1}{2} \sum_{r=1}^{2} \mathcal{L}_0^{\mathrm{pseudo}}(\widetilde{W}^{(0),r}) - \mathcal{L}_0^{\mathrm{pseudo}}(\theta^*) \right| \leq b_n$$

   *and*

$$\max_{1 \leq s \leq S} \left| \frac{1}{2} \sum_{r=1}^{2} \mathcal{L}_0^{\mathrm{pseudo}}(\widetilde{W}^{(0+s),r}) - \frac{1}{2} \sum_{r=1}^{2} \mathcal{L}_0^{\mathrm{pseudo}}(W^{(0+s),*,r}) \right| \leq b_n.$$

3. ***(Population separation)*** *For every $s \in \mathcal{A}_h$, $\mathcal{E}(s) \leq c_{\mathrm{gap}} \nu_n$, and for every $s \notin \mathcal{A}_h$, $\mathcal{E}(s) \geq (1 + c_{\mathrm{gap}}) \nu_n$.*

4. ***(Threshold stability)***

$$|\hat{\sigma} - \sigma_n| \leq b_n$$

   *and, for a sufficiently large constant $C_\Delta > 0$,*

$$c_{\mathrm{gap}} \nu_n + C_\Delta \epsilon_n < C_\tau(\sigma_n - b_n) \leq C_\tau(\sigma_n + b_n) < (1 + c_{\mathrm{gap}}) \nu_n - C_\Delta \epsilon_n.$$

**Proposition 2** (Verification of source-detection inputs). *Suppose Assumption 1 and Condition 7 hold, and the validation folds are independent of the fitted training-fold estimators conditional on the training folds. Then Condition 5 holds with probability tending to one.*

*Proof.* For each fold $r \in \{1, 2\}$, condition on the training sample $X^{(0)} \setminus X^{(0)[r]}$, all source samples, and the fitted matrices $\{\widetilde{W}^{(0),r}\} \cup \{\widetilde{W}^{(0+s),r} : 1 \leq s \leq S\}$. The validation observations in $X^{(0)[r]}$ are independent of these fitted matrices. For any fitted matrix $\Theta$ in Condition 7(i), the validation pseudolikelihood contribution $\sum_{j=1}^{p} f\left(2X_{ij}^{(0)} \sum_{k \neq j} \Theta_{jk} X_{ik}^{(0)}\right)$ is bounded by $p \log(1 + \exp(B_v))$. Hoeffding's inequality applied conditionally for each fixed fold and fitted matrix, followed by a union bound over the two folds and the $S + 1$ fitted losses in each fold, implies that, with conditional probability at least $1 - C(S + 1)(S \vee p)^{-c}$,

$$\max_{1 \leq s \leq S} \left| \Delta_s - \left\{ \frac{1}{2} \sum_{r=1}^{2} \mathcal{L}_0^{\mathrm{pseudo}}(\widetilde{W}^{(0+s),r}) - \frac{1}{2} \sum_{r=1}^{2} \mathcal{L}_0^{\mathrm{pseudo}}(\widetilde{W}^{(0),r}) \right\} \right| \leq Cp\sqrt{\frac{\log(S \vee p)}{n_{\min}^{\mathrm{val}}}}.$$

Combining this inequality with Condition 7(ii) gives

$$\max_{1 \leq s \leq S} |\Delta_s - \mathcal{E}(s)| \leq C_\Delta \epsilon_n$$

for a sufficiently large constant $C_\Delta$. Condition 7(iii) gives the informative-source and non-informative-source parts of Condition 5. Condition 7(iv) gives

$$c_{\mathrm{gap}}\nu_n + C_\Delta \epsilon_n < C_\tau \hat{\sigma} < (1 + c_{\mathrm{gap}})\nu_n - C_\Delta \epsilon_n.$$

Combining the event in Condition 7 with the validation concentration event gives probability at least $1 - Cp^{-c} - C(S + 1)(S \vee p)^{-c}$. After increasing the numerical constant in the deviation bound, the exponent $c$ can be chosen larger than 1, and this probability tends to one. Because $\epsilon_n = o(\nu_n)$, the four parts of Condition 5 hold with probability tending to one. This completes the proof. □

Assumptions 2, 3, 4, and 5, together with Conditions 1, 2, 3, 4, and 5, combine standard conditions for high-dimensional Ising structure learning and folded-concave regularization with specific conditions required to control cross-domain transfer. Below we interpret these assumptions and conditions, discuss their validity, and clarify their roles in our theoretical analysis.

**Assumption 2 (sparsity and source-to-target proximity).** The sparsity scaling $s_j = o(n_0/\log p)$ is the usual regime in which nodewise logistic lasso can consistently estimate neighborhoods in Ising models (Ravikumar et al., 2010; Negahban et al., 2012). The proximity level $h_j$ quantifies how close an informative source is to the target in $\ell_1$. When sources are generated by perturbations of the target interactions of the order used in our simulations, $h_j$ has the corresponding order and transfer can reduce the target estimation error. When $h_j$ is larger, naive pooling can incur negative transfer. This assumption is imposed only for sources in the informative set $\mathcal{A}$; the full Trans-Ising algorithm attempts to identify such sources via loss-based screening.

**Assumption 3 (information comparability across sources).** This condition requires the local curvature (Hessian) of each informative source risk to be comparable to the pooled risk curvature within a neighborhood of $\theta^*_{\backslash j}$. This condition excludes sources whose conditional distributions are nearly deterministic or whose local Fisher information degenerates relative to the pooled task. In bounded-degree Ising models with parameters lying in a common bounded neighborhood, such local comparability is typically mild and ensures that the pooled population minimizer $w^*_{A,\backslash j}$ stays close to $\theta^*_{\backslash j}$ (Lemma 1).

**Condition 1 (pooled RSC on a shifted cone).** Restricted strong convexity (RSC) is the standard curvature condition used in high-dimensional M-estimation rates. The cone here is *shifted* because $w^*_{A,\backslash j}$ is not assumed to be exactly supported on $S_j$; instead, its off-support mass $\|(w^*_{A,\backslash j})_{S_j^c}\|_1$ acts as an approximation error. Lemma 1 implies this mass is controlled by $h_j$. The shift is of order $h_j$ when informative sources are close to the target.

**Condition 2 (target RSC for bias correction and effective sparsity).** Step 2 estimates a correction $\delta$ using only target data, and its analysis requires target curvature. The index budget $t_j = \lceil C_h h_j/\lambda_\delta \rceil$ bounds the *effective sparsity* of $\delta^*_{\backslash j}$: since we only assume $\|\delta^*_{\backslash j}\|_1 \lesssim h_j$, the set of coordinates larger than $\lambda_\delta$ can have size at most on the order of $h_j/\lambda_\delta$. This means that the RSC condition is only needed on cones associated with sets $T$ of size $\leq t_j$, which is weaker when $h_j/\lambda_\delta$ is of smaller order than $s_j$. Lemma 5 uses the lower-curvature constant $\kappa_0 > 0$ together with the tolerance term in Condition 2. A stronger bound such as $\kappa_0 > 1/(a - 1)$ is sufficient for analyses that combine the SCAD concavity term with the target curvature, but it is not needed for Lemma 5.

**Assumption 4 (bounded nodewise fields and non-degenerate curvature).** Assumption 4 imposes a uniform bound on the nodewise linear predictors (equivalently, a row-wise $\ell_1$ bound such as $2\|\theta^{(r)}_{j,\backslash j}\|_1 \leq M$ for all $j$ and $r$). This condition holds, for example, under bounded degree together with uniformly bounded edge weights, but it is more general and directly controls the range of the logistic argument. It prevents near-separation and ensures that the logistic curvature $f''(\cdot)$ is bounded away from zero on the relevant neighborhood, supporting local RSC and concentration arguments (Ravikumar et al., 2010; Negahban et al., 2012).

**Condition 3 (computable stationary local solution in Step 2).** Because Step 2 is nonconvex and nonsmooth, global optimality is generally intractable. We therefore analyze a practically computable solution returned by an optimization routine (e.g., LLA) that achieves approximate stationarity (KKT residual control) and lies in a local basin around $\delta^*_{\backslash j}$. This is a common way to obtain theoretical guarantees for folded-concave penalties: the statistical analysis proceeds on a high-probability event where such a local solution exists and is reached from a reasonable initialization (e.g., initial estimator at $\delta = 0$).

**Assumption 5 (tuning and beta-min for exact selection).** The tuning sequences $\lambda_w \asymp \sqrt{\log p/N}$ and $\lambda_\delta, \lambda \asymp \sqrt{\log p/n_0}$ match the usual stochastic fluctuation scales in the pooled and target-only objectives. The separation $c_\lambda > c_\delta$ is needed to ensure that, on null coordinates, the SCAD subgradient in its linear region can dominate the $\ell_1$ correction penalty and the optimization tolerance, enabling false-positive control in Theorem 2. Finally, the beta-min condition ensures true nonzero interactions are large enough to survive the combined estimation error and the SCAD transition region (in particular, exceeding the SCAD threshold $a\lambda$), which is the standard condition for exact support recovery.

**Condition 5 (separability for consistent source detection).** Condition 5 ensures that the "signal" distinguishing informative sources from non-informative ones dominates the "noise" induced by finite-sample stochastic fluctuations. Specifically, it requires that the gap in population excess risk between the true source set $\mathcal{A}_h$ and heterogeneous sources is sufficiently large relative to the concentration rate of the empirical pseudolikelihood. This condition guarantees that the cross-validation based screening rule can consistently identify $\mathcal{A}_h$ with high probability.

## S.4. Lemmas for Section 4

**Lemma 1** (Implication for pooled-to-target bias). *Under Assumptions 2 and 3, there exists a constant $C > 0$ such that*

$$\|\delta^*_{\backslash j}\|_1 = \|\theta^*_{\backslash j} - w^*_{A,\backslash j}\|_1 \leq C\, h_j.$$

Lemma 1 bounds the *population* pooled-to-target discrepancy by $\|\delta^*_{\backslash j}\|_1 \lesssim h_j$. That is, the pooled population optimum cannot drift far from the target when the informative sources are $\ell_1$-close. When $h_j$ is large, this bound becomes loose, which corresponds to the negative-transfer regime.

**Lemma 2** (Uniform max-norm bound for logistic Hessians). *For the signed-design logistic loss $\ell^{(r)}_j(u) = \frac{1}{n_r}\sum_{i=1}^{n_r} f(z^{(r)\top}_i u)$ with $z^{(r)}_{i,\ell} \in \{-2, 2\}$, we have for all $u \in \mathbb{R}^{p-1}$, $\|\nabla^2 \ell^{(r)}_j(u)\|_{\max} \leq 1$.*

*This implies that, for all $u, v \in \mathbb{R}^{p-1}$,*

$$\|\nabla \ell^{(r)}_j(u) - \nabla \ell^{(r)}_j(v)\|_\infty \leq \|u - v\|_1.$$

*The same bounds hold for $\ell_{0,j}$, $\ell_{A,j}$ and the population risks $\mathcal{L}_{r,j}$ and $\mathcal{L}_{A,j}$.*

**Lemma 3** (Score concentration). *Under Assumptions 1 and 4, there exist constants $c_A, c_0, c > 0$ such that, with probability at least $1 - p^{-c}$,*

$$\|\nabla \ell_{A,j}(w^*_{A,\backslash j})\|_\infty \leq c_A \sqrt{\frac{\log p}{N}}, \qquad \|\nabla \ell_{0,j}(\theta^*_{\backslash j})\|_\infty \leq c_0 \sqrt{\frac{\log p}{n_0}}.$$

**Lemma 4** (Step 1 error). *Under Assumptions 2, 3, 4, Condition 1, and Lemma 3, with $\lambda_w = C_w\sqrt{(\log p)/N}$ for a sufficiently large absolute constant $C_w > 0$ and*

$$s_j \frac{\log p}{N} + \frac{\log p}{N} h_j^2 = o(1),$$

*the Step 1 estimator satisfies, w.h.p.,*

$$\|\hat{w}^A_{\backslash j} - w^*_{A,\backslash j}\|_2 \lesssim \sqrt{\frac{s_j \log p}{N}} + \left(\sqrt{\lambda_w \|(w^*_{A,\backslash j})_{S^c_j}\|_1} \wedge \|(w^*_{A,\backslash j})_{S^c_j}\|_1\right),$$

$$\|\hat{w}^A_{\backslash j} - w^*_{A,\backslash j}\|_1 \lesssim s_j \sqrt{\frac{\log p}{N}} + \|(w^*_{A,\backslash j})_{S^c_j}\|_1.$$

*In particular, Lemma 1 gives $\|(w^*_{A,\backslash j})_{S^c_j}\|_1 \lesssim h_j$, which implies*

$$\|\hat{w}^A_{\backslash j} - w^*_{A,\backslash j}\|_2 \lesssim \sqrt{\frac{s_j \log p}{N}} + \left([(\tfrac{\log p}{N})^{1/4} h_j^{1/2}] \wedge h_j\right), \qquad \|\hat{w}^A_{\backslash j} - w^*_{A,\backslash j}\|_1 \lesssim s_j \sqrt{\frac{\log p}{N}} + h_j.$$

Lemma 4 yields a Step 1 error decomposition into a variance term $\sqrt{s_j \log p/N}$ that depends on the total informative sample size $N$, and a heterogeneity/approximation term controlled by $\|(w^*_{A,\backslash j})_{S^c_j}\|_1 \lesssim h_j$ (Lemma 1). This shows that pooling stabilizes the initializer when $N \gg n_0$ and $h_j$ is of smaller order than the variance reduction.

**Lemma 5** (Step 2 error). *Suppose Assumptions 1, 2, 3, and 4, and Conditions 1, 2, 3, and 4 hold. Choose tuning levels*

$$\lambda_\delta = C_\delta \sqrt{\frac{\log p}{n_0}}, \qquad \lambda = C_\lambda \sqrt{\frac{\log p}{n_0}},$$

*where $\lambda$ is the SCAD penalty level in (18) and $C_\delta, C_\lambda > 0$ are sufficiently large absolute constants. Assume in addition that $\log p / n_0 = o(1)$, $s_j \sqrt{\log p/N} + h_j \lesssim \lambda_\delta$, and the Step 1 estimate satisfies, w.h.p.,*

$$\|\hat{w}_{\backslash j}^A - w_{A,\backslash j}^*\|_1 \lesssim \lambda_\delta.$$

*Let $\hat{\delta}_{\backslash j}^A$ be the stationary local minimizer returned by Step 2 satisfying Condition 3. Then, with high probability,*

$$\|\hat{\delta}_{\backslash j}^A - \delta_{\backslash j}^*\|_2 \lesssim \left( \left[ (\tfrac{\log p}{n_0})^{1/4} h_j^{1/2} \right] \wedge h_j \right), \quad \|\hat{\delta}_{\backslash j}^A - \delta_{\backslash j}^*\|_1 \lesssim h_j.$$

Lemma 5 shows that Step 2 estimates the cross-domain correction $\delta_{\backslash j}^*$ using only target data; its rates depend on $(n_0, p)$ and the heterogeneity scale $h_j$. The neighborhood sparsity $s_j$ does not enter the Step 2 rate above. When $h_j$ is not of the same order as $\sqrt{\log p / n_0}$ or smaller, the condition $s_j \sqrt{\log p/N} + h_j \lesssim \lambda_\delta$ can fail, and Step 2 no longer gives the rate in Lemma 5.

## S.5. Additional Details

### S.5.1. SCAD penalty

We use the Smoothly Clipped Absolute Deviation (SCAD) penalty (Fan & Li, 2001):

$$P_\lambda(\theta) = \begin{cases} \lambda|\theta|, & \text{if } |\theta| \le \lambda, \\ \dfrac{-\theta^2 + 2a\lambda|\theta| - \lambda^2}{2(a-1)}, & \lambda < |\theta| \le a\lambda, \quad (a > 2). \\ \dfrac{(a+1)\lambda^2}{2}, & \text{if } |\theta| > a\lambda, \end{cases} \tag{20}$$

### S.5.2. Remarks and Corollaries for Section 4

**Remark 3.** *Conditions 1–2 are theorem-local empirical curvature inputs. Proposition 1 verifies them from bounded-design restricted-eigenvalue conditions and empirical Gram concentration.*

**Remark 4** (On the Step 2 rate). *The mixed rate in Lemma 5 is characteristic of two-step transfer estimators for generalized linear models: it arises from the interaction between the curvature of the logistic loss and the heterogeneity scale $h_j$. A detailed argument follows the proof strategy in Tian & Feng (2023, Appendix), adapted to the nodewise Ising logistic loss.*

**Remark 5** (Selection under small heterogeneity). *The support-recovery proof applies under the rate and separation conditions in Theorem 2, including $r_{n,j} \le \lambda$ and the beta-min condition. If these conditions fail, Theorem 2 does not imply exact neighborhood recovery.*

**Remark 6** (Uniform versions). *Theorems 1–2 are stated for a fixed node $j$. If the corresponding fixed-node failure probabilities are $o(p^{-1})$ uniformly over $j = 1, \ldots, p$, then uniform results follow by a union bound. Alternatively, uniform versions may be proved directly by standard maximal inequalities under the corresponding uniform empirical curvature and score-control conditions, as in Li et al. (2022); Tian & Feng (2023).*

**Corollary 2** (Graph recovery after AND symmetrization). *Suppose the conclusion of Theorem 2 holds uniformly for all nodes $j = 1, \ldots, p$. Let $\hat{E}$ be the edge set obtained by the AND rule applied to the asymmetric nodewise estimates. Then $\mathbb{P}(\hat{E} = E) \to 1$.*

**Remark 7** (Why SCAD enables selection under weaker design assumptions than Lasso). *Classical $\ell_1$-penalized estimators (Lasso) typically require an irrepresentable / mutual incoherence condition on the population Hessian (e.g., $\|H_{S^c S} H_{SS}^{-1}\|_\infty \le 1 - \gamma$) to guarantee exact support recovery (Zhao & Yu, 2006; Wainwright, 2009; Bühlmann & van de Geer, 2011). In contrast, Theorem 2 establishes selection consistency under the target RSC condition (Condition 2), beta-min condition (Assumption 5), separation condition, Step 2 local-solution condition, and Step 2 cone condition, without imposing any irrepresentable-type constraints. This is consistent with the oracle property of folded-concave penalties such as SCAD/MCP, which reduce shrinkage for sufficiently large signals.*

### S.5.3. Explicit conditions for Theorem 2

For completeness, we restate the explicit sufficient conditions used for selection consistency. Let $r_{n,j}$ denote the nodewise error bound in Theorem 1. In Theorem 2, we require:

1. (Small-error regime for SCAD linear part) the nodewise error bound $r_{n,j}$ in Theorem 1 satisfies $r_{n,j} \leq \lambda$, which implies that, for any $k \in S_j^c$, we have $|\hat{\theta}_{jk}| \leq \lambda$ on the high-probability event.

2. (Separation between SCAD and correction penalty) there exists a sufficiently large universal constant $C_{\text{sel}} > 0$ such that

$$\lambda - \lambda_\delta - \varepsilon_{n,j} \geq C_{\text{sel}} \left( \sqrt{\frac{\log p}{n_0}} + s_j \sqrt{\frac{\log p}{N}} + h_j \right). \tag{21}$$

3. (Beta-min) Assumption 5 holds; that is,

$$|\theta_{jk}^*| \geq a\lambda + r_{n,j} \qquad \text{for all } k \in S_j.$$

## S.6. Proofs

### S.6.1. Proof of Lemma 1

*Proof.* By definition, $w_{A,\backslash j}^*$ satisfies the population first-order condition $\nabla \mathcal{L}_{A,j}(w_{A,\backslash j}^*) = 0$, i.e.,

$$0 = \sum_{r \in \{0\} \cup \mathcal{A}} \alpha_r \nabla \mathcal{L}_{r,j}(w_{A,\backslash j}^*).$$

For the target domain $r = 0$, $\theta_{\backslash j}^*$ minimizes $\mathcal{L}_{0,j}$, which implies $\nabla \mathcal{L}_{0,j}(\theta_{\backslash j}^*) = 0$. For each source $s \in \mathcal{A}$, let $w_{\backslash j}^{(s)}$ denote the nodewise parameter, where $\nabla \mathcal{L}_{s,j}(w_{\backslash j}^{(s)}) = 0$. By the integral form of Taylor's theorem,

$$\nabla \mathcal{L}_{s,j}(\theta_{\backslash j}^*) = \bar{H}_s(\theta_{\backslash j}^* - w_{\backslash j}^{(s)}), \qquad \bar{H}_s := \int_0^1 \nabla^2 \mathcal{L}_{s,j}\left( w_{\backslash j}^{(s)} + t(\theta_{\backslash j}^* - w_{\backslash j}^{(s)}) \right) dt.$$

Similarly, define

$$\bar{H}_A := \int_0^1 \nabla^2 \mathcal{L}_{A,j}\left( \theta_{\backslash j}^* + t(w_{A,\backslash j}^* - \theta_{\backslash j}^*) \right) dt.$$

Then

$$0 = \nabla \mathcal{L}_{A,j}(w_{A,\backslash j}^*) = \nabla \mathcal{L}_{A,j}(\theta_{\backslash j}^*) + \bar{H}_A(w_{A,\backslash j}^* - \theta_{\backslash j}^*)$$

Combining the above and using $\nabla \mathcal{L}_{0,j}(\theta_{\backslash j}^*) = 0$ gives

$$w_{A,\backslash j}^* - \theta_{\backslash j}^* = -\bar{H}_A^{-1} \times \sum_{s \in \mathcal{A}} \alpha_s \bar{H}_s (\theta_{\backslash j}^* - w_{\backslash j}^{(s)}).$$

Taking induced $\ell_1$ norms and using Assumption 3 gives

$$\|w_{A,\backslash j}^* - \theta_{\backslash j}^*\|_1 \leq C \sum_{s \in \mathcal{A}} \alpha_s \|\theta_{\backslash j}^* - w_{\backslash j}^{(s)}\|_1 \leq C h_j.$$

This completes the proof. □

### S.6.2. Proof of Lemma 2

*Proof.* For any coordinates $a, b$,

$$\left[\nabla^2 \ell_j^{(r)}(u)\right]_{ab} = \frac{1}{n_r} \sum_{i=1}^{n_r} f''(z_i^{(r)\top} u) \, z_{i,a}^{(r)} z_{i,b}^{(r)}.$$

Because $0 \leq f''(t) \leq 1/4$ for all $t$ and $|z_{i,a}^{(r)} z_{i,b}^{(r)}| \leq 4$, we have $\left|\left[\nabla^2 \ell_j^{(r)}(u)\right]_{ab}\right| \leq 1$ for all $u$. This gives $\|\nabla^2 \ell_j^{(r)}(u)\|_{\max} \leq 1$.

For the Lipschitz bound, use the integral form $\nabla \ell_j^{(r)}(u) - \nabla \ell_j^{(r)}(v) = \{\int_0^1 \nabla^2 \ell_j^{(r)}(v + t(u-v))\, dt\}(u-v)$, and then

$$\|\nabla \ell_j^{(r)}(u) - \nabla \ell_j^{(r)}(v)\|_\infty \leq \sup_{0 \leq t \leq 1} \|\nabla^2 \ell_j^{(r)}(v + t(u-v))\|_{\max} \|u-v\|_1 \leq \|u-v\|_1.$$

The population bounds follow by taking expectation of the entrywise bound. This completes the proof. $\qquad\square$

### S.6.3. Proof of Lemma 3

*Proof.* Fix a node $j$. For the target sample $X^{(0)}$, recall the signed design representation

$$Z_{\backslash j}^{(0)} = 2 \operatorname{diag}(x_{\cdot j}^{(0)})\, X_{\backslash j}^{(0)}, \qquad u_{j,i}(\vartheta) = [Z_{\backslash j}^{(0)}]_{i,\cdot}\, \vartheta, \qquad \widehat{p}_{j,i}(\vartheta) = \sigma(u_{j,i}(\vartheta)),$$

where $\sigma(u) = (1 + e^{-u})^{-1}$. Note that $x_{ik} \in \{-1, 1\}$ implies $|[Z_{\backslash j}^{(0)}]_{ik}| \leq 2$ for all $i, k$.

**(a) Target score at $\theta_{\backslash j}^*$.** For a fixed coordinate $k \neq j$, the $k$-th component of the target score is

$$\left[\nabla \ell_{0,j}(\theta_{\backslash j}^*)\right]_k = \frac{1}{n_0} \sum_{i=1}^{n_0} \xi_{ik}, \qquad \xi_{ik} := [Z_{\backslash j}^{(0)}]_{ik}\left(\widehat{p}_{j,i}(\theta_{\backslash j}^*) - 1\right).$$

We claim $\mathbb{E}[\xi_{ik} \mid X_{i,\backslash j}^{(0)}] = 0$. Indeed, set $\eta_i := 2 \sum_{m \neq j} \theta_{jm}^* x_{im}^{(0)}$. Then

$$u_{j,i}(\theta_{\backslash j}^*) = 2 x_{ij}^{(0)} \sum_{m \neq j} \theta_{jm}^* x_{im}^{(0)} = x_{ij}^{(0)} \eta_i.$$

Write $p_i := \sigma(\eta_i) = \mathbb{P}\{X_{ij}^{(0)} = 1 \mid X_{i,\backslash j}^{(0)}\}$ under the Ising conditional model. Then $\mathbb{P}\{X_{ij}^{(0)} = -1 \mid X_{i,\backslash j}^{(0)}\} = 1 - p_i = \sigma(-\eta_i)$. A short calculation gives

$$\mathbb{E}\left[x_{ij}^{(0)}\left(\sigma(x_{ij}^{(0)} \eta_i) - 1\right) \,\middle|\, X_{i,\backslash j}^{(0)}\right] = p_i\,(p_i - 1) + (1 - p_i)\, p_i = 0,$$

and multiplying by $2 x_{ik}^{(0)}$ yields $\mathbb{E}[\xi_{ik} \mid X_{i,\backslash j}^{(0)}] = 0$.

In addition, $|\widehat{p}_{j,i}(\theta_{\backslash j}^*) - 1| \leq 1$ and $|[Z_{\backslash j}^{(0)}]_{ik}| \leq 2$, which implies $|\xi_{ik}| \leq 2$ almost surely. By Hoeffding's inequality, for any $t > 0$,

$$\mathbb{P}\left(\left|\frac{1}{n_0} \sum_{i=1}^{n_0} \xi_{ik}\right| \geq t\right) \leq 2 \exp\left(-\frac{n_0 t^2}{8}\right).$$

Taking a union bound over $k \in V \setminus \{j\}$ and choosing $t = C\sqrt{\log p / n_0}$ gives

$$\mathbb{P}\left(\|\nabla \ell_{0,j}(\theta_{\backslash j}^*)\|_\infty \geq C\sqrt{\frac{\log p}{n_0}}\right) \leq 2 p^{-c}$$

for some $c > 0$.

**(b) Pooled score at $w_{A,\backslash j}^*$.** Recall the pooled objective

$$\ell_{A,j}(u) = \sum_{r \in \{0\} \cup \mathcal{A}} \alpha_r\, \ell_j^{(r)}(u),$$

where $\alpha_r = \frac{n_r}{N}$ and $N = \sum_{r \in \{0\} \cup \mathcal{A}} n_r$.

This gives

$$\nabla \ell_{A,j}(u) = \sum_{r \in \{0\} \cup \mathcal{A}} \alpha_r\, \nabla \ell_j^{(r)}(u), \qquad \nabla \ell_j^{(r)}(u) = \frac{1}{n_r} \sum_{i=1}^{n_r} \psi_i^{(r)}(u),$$

where $\psi_i^{(r)}(u) \in \mathbb{R}^{p-1}$ is the single-sample score contribution. At $u = w_{A,\backslash j}^*$, population optimality gives $\nabla\mathcal{L}_{A,j}(w_{A,\backslash j}^*) = 0$, which implies

$$\mathbb{E}\big[\nabla\ell_{A,j}(w_{A,\backslash j}^*)\big] = 0.$$

Fix $k \neq j$. The pooled score coordinate can be written as

$$\big[\nabla\ell_{A,j}(w_{A,\backslash j}^*)\big]_k = \frac{1}{N} \sum_{r \in \{0\} \cup \mathcal{A}} \sum_{i=1}^{n_r} \zeta_{ik}^{(r)},$$

where $\zeta_{ik}^{(r)}$ are independent across $(r, i)$ and uniformly bounded by a universal constant, since $x \in \{-1, 1\}$. Note that $\zeta_{ik}^{(r)}$ need not be mean-zero for each $r$, but the pooled mean equals zero:

$$\mathbb{E}\big[\nabla\ell_{A,j}(w_{A,\backslash j}^*)\big] = \nabla\mathcal{L}_{A,j}(w_{A,\backslash j}^*) = 0.$$

Define the centered variables

$$\tilde{\zeta}_{ik}^{(r)} := \zeta_{ik}^{(r)} - \mathbb{E}[\zeta_{ik}^{(r)}],$$

Then $\mathbb{E}[\tilde{\zeta}_{ik}^{(r)}] = 0$ and $|\tilde{\zeta}_{ik}^{(r)}|$ is still uniformly bounded. Then

$$\big[\nabla\ell_{A,j}(w_{A,\backslash j}^*)\big]_k = \frac{1}{N} \sum_{r \in \{0\} \cup \mathcal{A}} \sum_{i=1}^{n_r} \tilde{\zeta}_{ik}^{(r)},$$

and Hoeffding's inequality yields

$$\mathbb{P}\Big(\big|\big[\nabla\ell_{A,j}(w_{A,\backslash j}^*)\big]_k\big| \geq t\Big) \leq 2\exp\big(-cNt^2\big),$$

for a universal $c > 0$. A union bound over $k$ gives

$$\mathbb{P}\left(\big\|\nabla\ell_{A,j}(w_{A,\backslash j}^*)\big\|_\infty \geq C\sqrt{\frac{\log p}{N}}\right) \leq 2p^{-c'}$$

for some $c' > 0$.

**(c) Pooled score at $\theta_{\backslash j}^*$.** Decompose

$$\nabla\ell_{A,j}(\theta_{\backslash j}^*) = \Big(\nabla\ell_{A,j}(\theta_{\backslash j}^*) - \mathbb{E}[\nabla\ell_{A,j}(\theta_{\backslash j}^*)]\Big) + \mathbb{E}[\nabla\ell_{A,j}(\theta_{\backslash j}^*)].$$

The fluctuation term is an average of $N$ independent bounded summands, which gives

$$\big\|\nabla\ell_{A,j}(\theta_{\backslash j}^*) - \mathbb{E}[\nabla\ell_{A,j}(\theta_{\backslash j}^*)]\big\|_\infty \lesssim \sqrt{\frac{\log p}{N}} \quad \text{w.h.p.}$$

For the bias term, note that $\mathbb{E}[\nabla\ell_{A,j}(u)] = \nabla\mathcal{L}_{A,j}(u)$ and $\nabla\mathcal{L}_{A,j}(w_{A,\backslash j}^*) = 0$. By the integral form of Taylor's theorem,

$$\nabla\mathcal{L}_{A,j}(\theta_{\backslash j}^*) = \left\{\int_0^1 \nabla^2\mathcal{L}_{A,j}\big(w_{A,\backslash j}^* + t(\theta_{\backslash j}^* - w_{A,\backslash j}^*)\big)\,dt\right\}(\theta_{\backslash j}^* - w_{A,\backslash j}^*)$$

and Lemma 2 bounds the integrated Hessian in max norm by 1, which gives

$$\big\|\mathbb{E}[\nabla\ell_{A,j}(\theta_{\backslash j}^*)]\big\|_\infty = \|\nabla\mathcal{L}_{A,j}(\theta_{\backslash j}^*)\|_\infty \leq \|\theta_{\backslash j}^* - w_{A,\backslash j}^*\|_1 = \|\delta_{\backslash j}^*\|_1 \lesssim h_j,$$

where the last step uses Lemma 1. Combining both parts yields

$$\big\|\nabla\ell_{A,j}(\theta_{\backslash j}^*)\big\|_\infty \lesssim \sqrt{\frac{\log p}{N}} + h_j \quad \text{w.h.p.}$$

This completes the proof. $\qquad\square$

### S.6.4. Proof of Lemma 4

*Proof.* Fix a node $j$ and abbreviate

$$\hat{w} := \hat{w}_{\backslash j}^A, \qquad w^* := w_{A,\backslash j}^*, \qquad \Delta := \hat{w} - w^*.$$

Let $S := S_j = \mathrm{supp}(\theta_{\backslash j}^*)$ and $s := |S|$. Note that $w^*$ need not be supported on $S$, but by Lemma 1,

$$\|w_{S^c}^*\|_1 \leq \|w^* - \theta_{\backslash j}^*\|_1 = \|\delta_{\backslash j}^*\|_1 \lesssim h_j. \tag{22}$$

**Step 1: Basic inequality.** By optimality of $\hat{w}$ for (17),

$$\ell_{A,j}(\hat{w}) + \lambda_w \|\hat{w}\|_1 \leq \ell_{A,j}(w^*) + \lambda_w \|w^*\|_1.$$

Rearrange:

$$\ell_{A,j}(\hat{w}) - \ell_{A,j}(w^*) \leq \lambda_w(\|w^*\|_1 - \|\hat{w}\|_1). \tag{23}$$

Also, by convexity of $\ell_{A,j}$,

$$\ell_{A,j}(\hat{w}) - \ell_{A,j}(w^*) \geq \langle \nabla \ell_{A,j}(w^*), \Delta \rangle. \tag{24}$$

Combining (23)–(24) gives

$$\langle \nabla \ell_{A,j}(w^*), \Delta \rangle \leq \lambda_w(\|w^*\|_1 - \|w^* + \Delta\|_1). \tag{25}$$

**Step 2: Control of the $\ell_1$-difference.** Using decomposability of the $\ell_1$ norm,

$$\|w^*\|_1 - \|w^* + \Delta\|_1 = \|w_S^*\|_1 + \|w_{S^c}^*\|_1 - \|w_S^* + \Delta_S\|_1 - \|w_{S^c}^* + \Delta_{S^c}\|_1 \leq \|\Delta_S\|_1 - \|\Delta_{S^c}\|_1 + 2\|w_{S^c}^*\|_1. \tag{26}$$

On the other hand,

$$\langle \nabla \ell_{A,j}(w^*), \Delta \rangle \geq -\|\nabla \ell_{A,j}(w^*)\|_\infty \|\Delta\|_1.$$

Plugging this and (26) into (25) yields

$$-\|\nabla \ell_{A,j}(w^*)\|_\infty (\|\Delta_S\|_1 + \|\Delta_{S^c}\|_1) \leq \lambda_w (\|\Delta_S\|_1 - \|\Delta_{S^c}\|_1 + 2\|w_{S^c}^*\|_1).$$

Choose $\lambda_w \geq 2\|\nabla \ell_{A,j}(w^*)\|_\infty$ (w.h.p. by Lemma 3 when $\lambda_w = C_w \sqrt{\log p/N}$ and $C_w > 0$ is sufficiently large). Then

$$\frac{\lambda_w}{2}\|\Delta_{S^c}\|_1 \leq \frac{3\lambda_w}{2}\|\Delta_S\|_1 + 2\lambda_w\|w_{S^c}^*\|_1,$$

i.e. the shifted cone condition

$$\|\Delta_{S^c}\|_1 \leq 3\|\Delta_S\|_1 + 4\|w_{S^c}^*\|_1. \tag{27}$$

In particular, using (22),

$$\|\Delta_{S^c}\|_1 \leq 3\|\Delta_S\|_1 + Ch_j \qquad \text{w.h.p.} \tag{28}$$

Condition 1 includes the required containment of the segment between $\hat{w}$ and $w^*$ in the local RSC neighborhood.

**Step 3: Apply pooled RSC.** By Condition 1, for $\vartheta$ on the segment between $\hat{w}$ and $w^*$,

$$\ell_{A,j}(\hat{w}) - \ell_{A,j}(w^*) - \langle \nabla \ell_{A,j}(w^*), \Delta \rangle \geq \frac{\kappa_A}{2}\|\Delta\|_2^2 - \tau_A \frac{\log p}{N}\|\Delta\|_1^2.$$

Combine this with (23):

$$\frac{\kappa_A}{2}\|\Delta\|_2^2 - \tau_A \frac{\log p}{N}\|\Delta\|_1^2 \leq \lambda_w(\|w^*\|_1 - \|\hat{w}\|_1) - \langle \nabla \ell_{A,j}(w^*), \Delta \rangle$$

$$\leq \lambda_w (\|\Delta_S\|_1 - \|\Delta_{S^c}\|_1 + 2\|w_{S^c}^*\|_1) + \|\nabla \ell_{A,j}(w^*)\|_\infty \|\Delta\|_1.$$

Using again $\lambda_w \geq 2\|\nabla \ell_{A,j}(w^*)\|_\infty$ and (27), the right-hand side is bounded by

$$\frac{3\lambda_w}{2}\|\Delta_S\|_1 + 2\lambda_w\|w_{S^c}^*\|_1 \leq \frac{3\lambda_w}{2}\sqrt{s}\|\Delta\|_2 + 2\lambda_w\|w_{S^c}^*\|_1.$$

Also from (27),

$$\|\Delta\|_1 = \|\Delta_S\|_1 + \|\Delta_{S^c}\|_1 \leq 4\|\Delta_S\|_1 + 4\|w_{S^c}^*\|_1 \leq 4\sqrt{s}\|\Delta\|_2 + 4\|w_{S^c}^*\|_1.$$

**Step 4: Conclude the rates.** Using the previous bounds and absorbing the RSC tolerance term under $s \log p / N + (\log p / N) h_j^2 = o(1)$, we arrive at an inequality of the form

$$\|\Delta\|_2^2 \lesssim \lambda_w \sqrt{s} \|\Delta\|_2 + \lambda_w \|w_{S^c}^*\|_1.$$

Solving this quadratic inequality first yields

$$\|\Delta\|_2 \lesssim \lambda_w \sqrt{s} + \sqrt{\lambda_w \|w_{S^c}^*\|_1}.$$

Let $B := \|w_{S^c}^*\|_1$. If $B \geq \lambda_w$, then $\sqrt{\lambda_w B} \leq B$. If $B < \lambda_w$ and $s \geq 1$, then $\sqrt{\lambda_w B} \leq \lambda_w \leq \lambda_w \sqrt{s}$, and this term is absorbed into the leading term. If $s = 0$, (27) gives $\|\Delta\|_1 \leq 4B$, which implies $\|\Delta\|_2 \leq 4B$. Combining these cases,

$$\|\Delta\|_2 \lesssim \lambda_w \sqrt{s} + \left( \sqrt{\lambda_w B} \wedge B \right).$$

In addition, the shifted cone bound gives

$$\|\Delta\|_1 \lesssim s \lambda_w + \|w_{S^c}^*\|_1.$$

With $\lambda_w = C_w \sqrt{(\log p)/N}$ and $\|w_{S^c}^*\|_1 \lesssim h_j$, this yields

$$\|\Delta\|_2 \lesssim \sqrt{\frac{s \log p}{N}} + \left( \left[ (\tfrac{\log p}{N})^{1/4} h_j^{1/2} \right] \wedge h_j \right), \qquad \|\Delta\|_1 \lesssim s \sqrt{\frac{\log p}{N}} + h_j.$$

This completes the proof. $\qquad\square$

### S.6.5. Proof of Lemma 5

*Proof.* Fix a node $j$. Write

$$\hat{\delta} := \hat{\delta}_{\backslash j}^A, \qquad \delta^* := \delta_{\backslash j}^*, \qquad v := \hat{\delta} - \delta^*, \qquad \hat{\theta} := \hat{w}_{\backslash j}^A + \hat{\delta}.$$

Recall the Step 2 objective

$$Q(\delta) := \ell_{0,j}(\hat{w}_{\backslash j}^A + \delta) + \lambda_\delta \|\delta\|_1 + \sum_{k \neq j} P_\lambda(\hat{w}_{jk}^A + \delta_{jk}),$$

where $P_\lambda$ is the SCAD penalty with parameter $a > 2$ and level $\lambda$. Let

$$\theta^\dagger := \hat{w}_{\backslash j}^A + \delta^*.$$

Note that $\theta^\dagger = \theta_{\backslash j}^* + \Delta_w$, where $\Delta_w := \hat{w}_{\backslash j}^A - w_{A, \backslash j}^*$ is the Step 1 error.

By Condition 3(ii), $\hat{\delta} = \delta^* + v$ satisfies $\|\hat{\delta} - \delta^*\|_1 \leq r_\delta$ and is locally optimal on the $\ell_1$-ball around $\delta^*$. Taking $\delta = \delta^*$ yields

$$Q(\delta^* + v) \leq Q(\delta^*). \tag{29}$$

**Step 1: Basic inequality and Taylor expansion.** Expanding (29) yields

$$\ell_j^{(0)}(\theta^\dagger + v) - \ell_j^{(0)}(\theta^\dagger) \leq \lambda_\delta \big( \|\delta^*\|_1 - \|\delta^* + v\|_1 \big) + \sum_{k \neq j} \Big( P_\lambda(\theta_k^\dagger) - P_\lambda(\theta_k^\dagger + v_k) \Big). \tag{30}$$

Let $\tilde{\theta}$ be a point on the segment between $\theta^\dagger$ and $\theta^\dagger + v$. Then

$$\ell_j^{(0)}(\theta^\dagger + v) - \ell_j^{(0)}(\theta^\dagger) = \langle \nabla \ell_j^{(0)}(\theta^\dagger), v \rangle + \frac{1}{2} v^\top \nabla^2 \ell_j^{(0)}(\tilde{\theta}) \, v. \tag{31}$$

Combining (30) and (31) gives

$$\langle \nabla \ell_j^{(0)}(\theta^\dagger), v \rangle + \frac{1}{2} v^\top \nabla^2 \ell_j^{(0)}(\tilde{\theta}) \, v \leq \lambda_\delta \big( \|\delta^*\|_1 - \|\delta^* + v\|_1 \big) + \sum_{k \neq j} \Big( P_\lambda(\theta_k^\dagger) - P_\lambda(\theta_k^\dagger + v_k) \Big). \tag{32}$$

For SCAD, the subgradient is uniformly bounded: for all $t$, any $g \in \partial P_\lambda(t)$ satisfies $|g| \leq \lambda$. Equivalently, $P_\lambda(\cdot)$ is $\lambda$-Lipschitz. This implies that, for any $u, b \in \mathbb{R}$,

$$P_\lambda(u) - P_\lambda(u + b) \leq \lambda|b|.$$

Applying this coordinate-wise yields the clean bound

$$\sum_{k \neq j} \left( P_\lambda(\theta_k^\dagger) - P_\lambda(\theta_k^\dagger + v_k) \right) \leq \lambda\|v\|_1. \tag{33}$$

**Step 2: Score control at $\theta^\dagger$.** Recall $\theta^\dagger = \theta_{\backslash j}^* + \Delta_w$ where $\Delta_w = \hat{w}_{\backslash j}^A - w_{A,\backslash j}^*$. By Lemma 2,

$$\|\nabla\ell_{0,j}(\theta^\dagger) - \nabla\ell_{0,j}(\theta_{\backslash j}^*)\|_\infty \leq \|\theta^\dagger - \theta_{\backslash j}^*\|_1 = \|\Delta_w\|_1.$$

This gives

$$\|\nabla\ell_{0,j}(\theta^\dagger)\|_\infty \leq \|\nabla\ell_{0,j}(\theta_{\backslash j}^*)\|_\infty + \|\Delta_w\|_1.$$

This implies

$$|\langle\nabla\ell_{0,j}(\theta^\dagger), v\rangle| \leq \left( \|\nabla\ell_{0,j}(\theta_{\backslash j}^*)\|_\infty + \|\Delta_w\|_1 \right) \|v\|_1.$$

By Lemma 3, $\|\nabla\ell_{0,j}(\theta_{\backslash j}^*)\|_\infty \lesssim \sqrt{\log p/n_0}$ w.h.p. The Step 1 hypothesis of this lemma gives $\|\Delta_w\|_1 \lesssim \lambda_\delta$ w.h.p. Because $\lambda_\delta \asymp \sqrt{\log p/n_0}$, this gives

$$|\langle\nabla\ell_{0,j}(\theta^\dagger), v\rangle| \lesssim \lambda_\delta \|v\|_1 \text{ w.h.p.} \tag{34}$$

**Step 3: $\ell_1$ term via effective sparsity of $\delta^*$.** The vector $\delta^*$ is not assumed sparse, but satisfies $\|\delta^*\|_1 \leq C_h h_j$. Define the "large" index set at level $\lambda_\delta$:

$$T := T(\lambda_\delta) := \{k \neq j : |\delta_k^*| > \lambda_\delta\}, \qquad t := |T|.$$

Then $t \leq \|\delta^*\|_1/\lambda_\delta \leq C_h h_j/\lambda_\delta$. By decomposability,

$$\|\delta^*\|_1 - \|\delta^* + v\|_1 \leq \|v_T\|_1 - \|v_{T^c}\|_1 + 2\|\delta_{T^c}^*\|_1 \leq \|v_T\|_1 - \|v_{T^c}\|_1 + 2C_h h_j. \tag{35}$$

**Step 4: Cone condition and mixed rate.** Condition 4 gives

$$\|v_{T^c}\|_1 \leq 3\|v_T\|_1 + 3C_h h_j.$$

Together with Condition 3, this permits the use of Condition 2 in the Taylor expansion. Combine (32), (33), (34), (35), and Condition 2. With high probability,

$$\frac{\kappa_0}{2}\|v\|_2^2 - \tau_0\frac{\log p}{n_0}\|v\|_1^2 \lesssim \lambda_\delta\|v\|_1 + \lambda\|v\|_1 + \lambda_\delta(\|v_T\|_1 - \|v_{T^c}\|_1) + \lambda_\delta h_j. \tag{36}$$

Choose $\lambda \asymp \lambda_\delta \asymp \sqrt{\log p/n_0}$. By Condition 4,

$$\|v\|_1 = \|v_T\|_1 + \|v_{T^c}\|_1 \leq 4\|v_T\|_1 + 3C_h h_j \lesssim \sqrt{t}\|v\|_2 + h_j.$$

Substituting this bound into (36) reduces the bound to an inequality in $\|v\|_2$:

$$\|v\|_2^2 \lesssim \lambda_\delta\sqrt{t}\|v\|_2 + \lambda_\delta h_j + \frac{\log p}{n_0}\left(\sqrt{t}\|v\|_2 + h_j\right)^2.$$

Using $t \lesssim h_j/\lambda_\delta$ and $\lambda_\delta \asymp \sqrt{\log p/n_0}$, put $a_n := \lambda_\delta$ and $x := \|v\|_2$. The preceding inequality gives

$$x^2 \lesssim a_n\sqrt{t}\,x + a_n h_j + a_n^2(\sqrt{t}\,x + h_j)^2.$$

Because $t \lesssim h_j/a_n$, $h_j \lesssim a_n$, and $a_n^2 = \log p/n_0 = o(1)$, the coefficient of $x^2$ in $a_n^2 t x^2$ is $O(a_n h_j) = o(1)$ and can be absorbed into the left-hand side. The remaining terms in $a_n^2(\sqrt{t}\,x + h_j)^2$ are bounded by a constant multiple of $(a_n h_j)^{1/2}x + a_n h_j$. This gives $x^2 \lesssim (a_n h_j)^{1/2}x + a_n h_j$. Solving this scalar inequality gives

$$\|v\|_2 \lesssim (a_n h_j)^{1/2} = \left(\frac{\log p}{n_0}\right)^{1/4} h_j^{1/2}.$$

In addition, the cone bound gives

$$\|v\|_1 \lesssim \sqrt{t}\|v\|_2 + h_j \lesssim h_j,$$

which implies $\|v\|_2 \leq \|v\|_1 \lesssim h_j$. Combining the two bounds,

$$\|v\|_2 \lesssim \left(\left[(\tfrac{\log p}{n_0})^{1/4}h_j^{1/2}\right] \wedge h_j\right), \qquad \|v\|_1 \lesssim h_j,$$

as claimed. This completes the proof. □

### S.6.6. Proof of Theorem 1

*Proof.* Fix a node $j$. Recall

$$\hat{\theta}_{\backslash j} = \hat{w}^A_{\backslash j} + \hat{\delta}^A_{\backslash j}, \qquad \theta^*_{\backslash j} = w^*_{A,\backslash j} + \delta^*_{\backslash j}.$$

Define the Step 1 error $\Delta_w := \hat{w}^A_{\backslash j} - w^*_{A,\backslash j}$ and the Step 2 error $v := \hat{\delta}^A_{\backslash j} - \delta^*_{\backslash j}$. We condition on the intersection of the event on which Conditions 1–4 hold, the score-concentration event in Lemma 3, and the high-probability events in Lemmas 4 and 5. This intersection has probability at least $1 - C_0 p^{-c_0}$ after changing constants. Lemma 4 and the rate condition $s_j\sqrt{\log p/N} + h_j \lesssim \lambda_\delta$ give

$$\|\Delta_w\|_1 \lesssim s_j\sqrt{\frac{\log p}{N}} + h_j \lesssim \lambda_\delta.$$

This verifies the Step 1 condition in Lemma 5. Then

$$\hat{\theta}_{\backslash j} - \theta^*_{\backslash j} = \Delta_w + v,$$

and the triangle inequality gives

$$\|\hat{\theta}_{\backslash j} - \theta^*_{\backslash j}\|_2 \leq \|\Delta_w\|_2 + \|v\|_2.$$

Apply Lemma 4 to bound $\|\Delta_w\|_2$ and Lemma 5 to bound $\|v\|_2$. Combining the two bounds yields the claimed $\ell_2$ rate. The same triangle inequality, together with the $\ell_1$ bounds in Lemmas 4 and 5, gives

$$\|\hat{\theta}_{\backslash j} - \theta^*_{\backslash j}\|_1 \lesssim s_j\sqrt{\frac{\log p}{N}} + h_j.$$

This completes the proof. □

### S.6.7. Proof of Theorem 2

*Proof.* Fix a node $j \in V$ and let $S := S_j = \{k \neq j : \theta^*_{jk} \neq 0\}$. Recall $\hat{\theta}_{\backslash j} = \hat{w}^A_{\backslash j} + \hat{\delta}^A_{\backslash j}$. By Theorem 1, with probability tending to one,

$$\|\hat{\theta}_{\backslash j} - \theta^*_{\backslash j}\|_\infty \leq \|\hat{\theta}_{\backslash j} - \theta^*_{\backslash j}\|_2 \leq r_{n,j}.$$

On this event, we prove the following two claims.

**(a) Sign Consistency and Vanishing Bias on $S$.** For any $k \in S$, under the beta-min condition in Theorem 2,

$$|\hat{\theta}_{jk} - \theta^*_{jk}| \leq r_{n,j} < |\theta^*_{jk}|,$$

which implies that $\text{sign}(\hat{\theta}_{jk}) = \text{sign}(\theta^*_{jk})$. In addition,

$$|\hat{\theta}_{jk}| \geq |\theta^*_{jk}| - \|\hat{\theta}_{\backslash j} - \theta^*_{\backslash j}\|_\infty \geq (a\lambda + r_{n,j}) - r_{n,j} = a\lambda.$$

This has two implications:

1. Since $|\hat{\theta}_{jk}| > 0$, we have $\hat{S}_j \supseteq S$ (No false negatives).

2. Since $|\hat{\theta}_{jk}| \geq a\lambda$, we are in the flat region of the SCAD penalty. The subgradient of the SCAD penalty is zero:

$$P'_\lambda(|\hat{\theta}_{jk}|) = 0 \quad \forall k \in S.$$

This property, namely zero penalty derivative for large coefficients, is specific to folded-concave penalties such as SCAD and distinguishes this proof from Lasso. For Lasso, the penalty derivative would be $\lambda \cdot \mathrm{sign}(\hat{\theta}_{jk})$, introducing a bias that requires the Irrepresentable Condition to manage.

**(b) No False Positives (Selection Consistency).** We show $\hat{\theta}_{jk} = 0$ for all $k \in S^c$. For $k \in S^c$, suppose for contradiction that $\hat{\theta}_{jk} \neq 0$. On this event and under $r_{n,j} \leq \lambda$, we have $0 < |\hat{\theta}_{jk}| \leq \lambda$. The SCAD subgradient with respect to $\hat{\theta}_{jk}$ has absolute value $\lambda$. By Condition 3, there exist $g_{jk} \in \partial|\hat{\delta}_{jk}^A|$ and $h_{jk} \in \partial P_\lambda(\hat{\theta}_{jk})$ such that

$$\left| \left[\nabla \ell_{0,j}(\hat{\theta}_{\backslash j})\right]_k + \lambda_\delta g_{jk} + h_{jk} \right| \leq \varepsilon_{n,j}.$$

Because $|g_{jk}| \leq 1$ and $|h_{jk}| = \lambda$, this implies

$$\left| \left[\nabla \ell_{0,j}(\hat{\theta}_{\backslash j})\right]_k \right| \geq \lambda - \lambda_\delta - \varepsilon_{n,j}.$$

The integral form of Taylor's theorem and Lemma 2 give

$$\left| \left[\nabla \ell_{0,j}(\hat{\theta}_{\backslash j})\right]_k \right| \leq \|\nabla \ell_{0,j}(\theta_{\backslash j}^*)\|_\infty + \|\hat{\theta}_{\backslash j} - \theta_{\backslash j}^*\|_1.$$

Lemma 3 and Theorem 1 imply

$$\left| \left[\nabla \ell_{0,j}(\hat{\theta}_{\backslash j})\right]_k \right| \lesssim \sqrt{\frac{\log p}{n_0}} + s_j\sqrt{\frac{\log p}{N}} + h_j.$$

This contradiction implies that $\hat{\theta}_{jk} = 0$ for all $k \in S^c$.

We have shown that $\hat{S}_j \supseteq S$ from beta-min and flat penalty region and $\hat{S}_j \subseteq S$ from noise control dominating the gradient. This gives $\hat{S}_j = S$. The result holds without imposing the mutual incoherence condition on the Hessian, because the SCAD derivative equals zero on the true support. $\square$

### S.6.8. Proof of Theorem 3

In this section, we prove Theorem 3 from the graph-level separation and threshold condition.

*Proof.* Recall that the compatibility score for source $s$ is the difference between the averaged empirical cross-validation losses on the target domain. Define

$$\bar{\mathcal{L}}_0 := \frac{1}{2}\sum_{r=1}^{2} \hat{\mathcal{L}}_0^{(r)}, \qquad \bar{\mathcal{L}}_s := \frac{1}{2}\sum_{r=1}^{2} \hat{\mathcal{L}}_s^{(r)}.$$

Then

$$\Delta_s = \bar{\mathcal{L}}_s - \bar{\mathcal{L}}_0 = \frac{1}{2}\sum_{r=1}^{2} \left( \hat{\mathcal{L}}_s^{(r)} - \hat{\mathcal{L}}_0^{(r)} \right).$$

On the event in Condition 5, we have

$$\max_{1 \leq s \leq S} |\Delta_s - \mathcal{E}(s)| \leq C\epsilon_n$$

and

$$c_{\mathrm{gap}}\nu_n + C\epsilon_n < C_\tau \hat{\sigma} < (1 + c_{\mathrm{gap}})\nu_n - C\epsilon_n.$$

If $s \in \mathcal{A}_h$, then $\mathcal{E}(s) \leq c_{\mathrm{gap}}\nu_n$, which implies

$$\Delta_s \leq \mathcal{E}(s) + C\epsilon_n \leq c_{\mathrm{gap}}\nu_n + C\epsilon_n < C_\tau \hat{\sigma}.$$

This gives $s \in \hat{\mathcal{A}}$. If $s \notin \mathcal{A}_h$, then $\mathcal{E}(s) \geq (1 + c_{\mathrm{gap}})\nu_n$, which implies

$$\Delta_s \geq \mathcal{E}(s) - C\epsilon_n \geq (1 + c_{\mathrm{gap}})\nu_n - C\epsilon_n > C_\tau \hat{\sigma}.$$

This gives $s \notin \hat{\mathcal{A}}$. The event in Condition 5 has probability tending to one, which proves $\mathbb{P}(\hat{\mathcal{A}} = \mathcal{A}_h) \to 1$. This completes the proof. $\qquad\square$

