# OpenReview forum: "Transfer Learning in High-dimensional Ising Models"
_ICML.cc/2026/Conference — ICML 2026 regular_

### Official Review · Reviewer_PrGy · 2026-03-06

**Soundness:** 3
**Presentation:** 3
**Significance:** 2
**Originality:** 2
**Overall Recommendation:** 4
**Confidence:** 4

**Summary:**

This paper studies transfer learning in high-dimensional Ising models, where a target dataset with limited samples is assisted by multiple auxiliary datasets from related but potentially different distributions.
The authors propose Trans-Ising, a two-stage framework that first identifies informative source datasets and obtains an initial pooled estimator, then performs bias-corrected estimation using a dual-regularization scheme combining $\ell_1$ and SCAD penalties.
The paper provides theoretical guarantees for estimation error and graph structure recovery, and demonstrates the effectiveness of the method through synthetic and real-data applications.

**Compliance With Llm Reviewing Policy:**

Affirmed.

**Final Justification:**

The rebuttal has addressed my main concerns. Newly added systematic sensitivity analysis is broad. Added complexity analysis provides a clearer understanding of the practical computational cost. I will increase my confidence, and keep positive assessment.

**Key Questions For Authors:**

Based on the weakness part, I summarize key questions as follows. I will increase my score if these are well addressed.
1. Could the authors discuss about computational complexity, or scalability to larger graphs over the proposed method?
2. Could the authors provide technical comparisons between the proposed approach and prior frameworks (like prior transfer-learning methods for Gaussian graphical models and high-dimensional GLMs mentioned in the paper)?
3. Could the authors provide a more systematic sensitivity analysis of how performance of the proposed method varies (or degrades) as the source-target discrepancy becomes large, or when the proposed transfer mechanism ceases to be effective?

**Limitations:**

- I suggest that the authors discuss about computational complexity, or scalability to larger graphs over the proposed method.
- I suggest that  the authors provide technical comparisons between the proposed approach and prior frameworks (like prior transfer-learning methods for Gaussian graphical models and high-dimensional GLMs mentioned in the paper).
- I suggest that authors provide a more systematic sensitivity analysis of how performance of the proposed method varies (or degrades) as the source-target discrepancy becomes large, or when the proposed transfer mechanism ceases to be effective.

**Strengths And Weaknesses:**

### Strengths
- This paper is well-motivated as there exist many practical scenarios where a target dataset is limited in scales, but can leverage multiple auxiliary datasets from related distributions. Also, existing transfer learning theory remain relatively scarce for Ising models.
- The theory is solid and logically coherent. Also, the authors interpret the proposed assumptions and discuss their validity in the theoretical analysis in Appendix.
- The data-driven source detection algorithm proposed in this paper filters out harmful sources, enhancing practical robustness to negative transfer.
- The paper provides various domains, Mutation Data (main text),  Online Transaction Analysis, and Movie data analysis (in appendix). These help strengthen the theory of the paper.
### Weaknesses
- The proposed method involves multiple components (source detection, etc.), but the paper provides little discussion on computational complexity or scalability to larger graphs. I only see the authors admit that `cross-validated screening can be costly when many auxiliaries are available`. A more explicit and detailed analysis of runtime and practical implementation considerations would strengthen the work.
- The paper positions it is motivated by prior transfer-learning methods (line 83, page 2), eg, Gaussian graphical models and high-dimensional GLMs.
However, the key technical comparisons between the proposed approach and these prior frameworks are not clearly clarified. It remains unclear whether this work proposes fundamentally new design elements for the Ising setting or mainly an incremental extension of existing transfer-learning ideas.
- The paper introduces a source detection mechanism to mitigate negative transfer, and the synthetic study varies perturbation levels between target and auxiliary domains. However, the experiments do not provide a more systematic sensitivity analysis of how performance of the proposed method varies (or degrades) as the source-target discrepancy becomes large, or when the proposed transfer mechanism ceases to be effective (in other words, the ability boundary of the proposed method).
Adding additional analyses could help better understand the ability boundary of the proposed method.

---

> ### Author Rebuttal · Authors · 2026-03-31
>
> We thank the reviewer for the constructive feedback and for clearly identifying the three issues that most needed strengthening. We address each concern below.
>
> # Q1: Computational complexity and scalability
>
> The computational cost of Trans-Ising is dominated by three components:
>
> - **Step 1** (pooled nodewise logistic lasso): $p$ independent neighborhood-selection problems are fitted on the pooled sample of size $N = n_0 + \sum_{s \in \mathcal{A}} n_s$, yielding overall complexity on the order of $O(p^2 N)$ up to solver- and cross-validation-dependent constants.
>
> - **Step 2** (target-only bias correction): $p$ independent correction problems are fitted on the target sample only, giving $O(p^2 n_0)$ up to solver- and tuning-dependent constants for the nonconvex correction step.
>
> - **Source detection**: with $F$ target folds, for each candidate source the method refits nodewise models on the target training folds with and without that source, leading to an additional cost that scales linearly in the number of candidate sources $S$ and roughly as $O(FSp^2(n_0+n_s))$ for equal-sized sources.
>
> Thus, the dominant cost is quadratic in the number of variables and linear in the relevant sample size, with an additional linear factor in the number of auxiliary datasets during screening. The estimation stage is naturally parallelizable across nodes, and the screening stage across candidate sources. In practice, this makes the method scalable to moderately large graph sizes (e.g., hundreds of nodes), while the main computational bottleneck for larger graphs is the repeated nodewise fitting in the source-screening stage. We will clarify this scaling behavior and parallelization structure in the revised manuscript.
>
> # Q2: Technical comparison with prior transfer-learning frameworks
>
> We agree that the paper should position itself more clearly. We state the key distinctions below:
>
> |**Aspect**|**Trans-CLIME**|**Trans-GLM**|**Ours**|
> |:-|:-|:-|:-|
> |Model|Gaussian graphical model|High-dimensional GLM|High-dimensional Ising model|
> |Estimating backbone|CLIME-type moment constraints|Penalized GLM likelihood|Nodewise logistic pseudolikelihood|
> |Unknown source relevance|Informative set assumed given|Yes|Yes|
> |Main correction step|Moment-equation transfer + aggregation|$\ell_1$-based target correction|$\ell_1$ correction + SCAD on updated coefficients|
> |Exact support recovery of final transferred estimator|Not the main result|Not established|Yes (Theorem 2)|
>
> In short, our framework provides exact support/sign recovery of the final transferred estimator via the dual-regularization correction step, and selective transfer under unknown source relevance in the Ising setting. We will make this positioning and comparison more explicit in the revision.
>
> # Q3: Systematic sensitivity analysis
>
> We thank the reviewer for this important and well-motivated suggestion. We have conducted a comprehensive sensitivity analysis varying all three key experimental axes—$\sigma$, $n_0$, and $n_s$—at $p = 200$ with 3 informative + 9 non-informative sources (estimation error measured by relative Frobenius error, averaged over 50 replications). The key findings are as follows:
>
> **(a) Source-target discrepancy** ($n_0=160$; $n_s=300$):
>
> |$\sigma$|Oracle|Trans|Naive|Pooled|
> |:-:|:-:|:-:|:-:|:-:|
> |0.01|0.433|0.433|0.891|0.666|
> |0.10|0.441|0.442|0.890|0.668|
> |0.20|0.471|0.471|0.891|0.687|
> |0.50|0.754|0.829|0.891|0.849|
> |1.00|0.979|0.891|0.891|0.952|
>
> In our current setup, transfer
> is strongly effective through about $\sigma=0.2$, begins to degrade
> at $\sigma=0.5$, and by $\sigma=1.0$ the screening step
> essentially switches transfer off and returns to the target-only
> baseline. This is consistent with the intended robustness of the source detector.
>
> **(b) Target sample size** ($n_s=300$; $\sigma=0.1$):
>
> |$n_0$|Oracle|Trans|Naive|Pooled|
> |:-:|:-:|:-:|:-:|:-:|
> |80 |0.461|0.461|0.966|0.758|
> |120|0.449|0.448|0.929|0.709|
> |160|0.441|0.441|0.889|0.673|
> |300|0.427|0.427|0.789|0.572|
>
> The benefit is largest when the target sample is smallest, which is the regime where transfer learning is most needed.
> As $n_0$ grows, the target-only baseline becomes stronger and the relative transfer gain narrows, but the transferred estimator remains better throughout the sweep.
>
> **(c) Per-source auxiliary sample size** ($n_0=160$; $\sigma=0.1$):
>
> |$n_s$|Oracle|Trans|Naive|Pooled|
> |:-:|:-:|:-:|:-:|:-:|
> |100|0.577|0.577|0.891|0.724|
> |200|0.487|0.486|0.892|0.690|
> |300|0.441|0.441|0.891|0.672|
> |500|0.397|0.397|0.890|0.653|
>
> As informative auxiliary sample size increases, transfer improves monotonically.
> The main practical message is therefore: Trans-Ising helps most when the target sample is small,
> the informative auxiliaries are not too far from the target,
> and the screened informative sources contribute enough additional samples for the variance-reduction benefit to dominate the heterogeneity terms.
> We will include these sensitivity results in the revised paper.

---

> > ### Author Rebuttal · Reviewer_PrGy · 2026-04-03
> >
> > Thank you for the detailed and well-structured response. The rebuttal has addressed my main concerns. Newly added systematic sensitivity analysis is broad. Added complexity analysis provides a clearer understanding of the practical computational cost. I will increase my confidence, and keep positive assessment.

---

> > > ### Author Response · Authors · 2026-04-03
> > >
> > > We sincerely thank the reviewer for the thoughtful reassessment and for the generous feedback. The three key suggestions raised in the original review — computational complexity analysis, technical comparison with prior frameworks, and systematic sensitivity analysis — were instrumental in strengthening our paper, and we are grateful for the guidance. We will ensure all additions are clearly presented in the revised manuscript.

---

### Official Review · Reviewer_vyew · 2026-03-13

**Soundness:** 3
**Presentation:** 3
**Significance:** 1
**Originality:** 1
**Overall Recommendation:** 1
**Confidence:** 5

**Summary:**

This paper studies transfer learning for high-dimensional Ising models, where the goal is to estimate the target interaction network from a small target sample together with multiple auxiliary datasets of unknown relevance. The authors propose Trans-Ising, a two-stage procedure tailored to discrete graphical models. In the first stage, the method pools the target data with selected auxiliary sources and fits nodewise $\ell_1$-regularized logistic regressions to obtain an initial estimator. In the second stage, it performs a target-only bias-correction step that combines an $\ell_1$ penalty on the correction term with a folded-concave SCAD penalty on the updated coefficients, with the aim of improving edge recovery and reducing the shrinkage bias of standard lasso-based estimators. Since auxiliary datasets may be uninformative and can cause negative transfer, the paper further introduces a data-driven source screening rule based on held-out target pseudolikelihood loss, which selects sources whose inclusion improves or does not substantially worsen target-domain fit. On the theoretical side, the paper establishes estimation error bounds for the oracle version of the method, proves support recovery and sign consistency for the final estimator, and shows consistency of the source detection procedure under suitable high-dimensional conditions. Empirically, the method is evaluated on synthetic Ising models with different graph structures and on cancer mutation data, where it consistently improves over target-only estimation and naive pooling, especially when informative auxiliary sources are available.

**Compliance With Llm Reviewing Policy:**

Affirmed.

**Final Justification:**

I appreciate the authors' thoughtful rebuttal and their efforts to clarify several technical points. The paper is competently executed and reasonably well presented, and the additional discussion helps address some issues of exposition and empirical interpretation. However, the rebuttal did not change my main concern, which is originality. My assessment remains that the paper is too close in overall paradigm to prior transfer learning work for high-dimensional GLMs, while the core Ising-model estimation machinery is itself classical. More importantly, the rebuttal did not sufficiently address the existence of closely related prior work on meta-learning for high-dimensional Ising models, which further weakens the claimed novelty. Overall, this research's central aspect comprises a careful adaptation of existing ideas, but I remain unconvinced that the marginal conceptual contribution is strong enough for acceptance at ICML.

**Key Questions For Authors:**

1. The overall paradigm of the paper appears very close to the transfer-learning framework of Tian and Feng (2023) [1] for high-dimensional GLMs. Could the authors clarify what the substantive methodological differences are beyond specializing the framework to the Ising-model setting? In particular, which parts of the procedure, analysis, or guarantees are genuinely new and specific to Ising models, rather than parallel adaptations of the existing transfer-GLM paradigm?

2. The paper also seems closely related to prior work on meta-learning for high-dimensional Ising model selection [2]. Could the authors explain in detail the substantive differences between their work and the meta-Ising formulation, both at the levels of problem setup and technical development? More importantly, why do the authors believe these differences are sufficient to support a strong novelty claim, given that both works use auxiliary Ising tasks, nodewise $\\ell_1$-regularized logistic regression, and establish related support-recovery/sample-complexity-type guarantees?

3. In the simulation study, the target sample size, ambient dimension, and auxiliary sample sizes appear to be fixed at a single main regime (e.g., $n_0=160$, $p=200$, $S=12$, $n_s=300$), while the experiments mainly vary the number of informative sources and the perturbation level. Could the authors clarify how robust the empirical conclusions are to other regimes, especially more challenging ones with smaller $n_0$, larger $p/n_0$, fewer auxiliary samples per source, or weaker separation between informative and non-informative sources? Since the main motivation of the paper is high-dimensional transfer under limited target samples, I would find it important to see whether the claimed advantages persist beyond the particular simulation settings currently considered.

[1] Tian, Y. and Y. Feng (2023). Transfer learning under high-dimensional generalized linear models. *Journal of the American Statistical Association* 118 (544), 2684–2697.

[2] Xie, H. and J. Honorio (2024). Meta learning for support recovery of high-dimensional Ising models. *Transactions on Machine Learning Research*.

**Limitations:**

Yes

**Strengths And Weaknesses:**

The paper is technically solid and generally well presented. The proposed procedure is clearly described, the two-stage estimation and source-detection components are implemented with care, and the empirical results are broadly consistent with the claimed benefits of leveraging auxiliary data. In this sense, I find the work reasonably sound and easy to follow.

That said, I have substantial concerns about its significance and originality. At a high level, the paper appears to take the high-dimensional transfer-learning framework for generalized linear models of Tian and Feng (2023) [1] and instantiate it for the Ising-model setting in a largely parallel way, rather than introducing a fundamentally new transfer-learning principle. Moreover, the statistical foundations for high-dimensional Ising model selection via nodewise $\\ell_1$-regularized logistic regression are already classical and well developed (e.g. [2]), so the underlying model class and much of the technical machinery are not new.

More importantly, the paper seems to miss a highly relevant meta-Ising work ([3]) that is very close in spirit and structure. The similarity is not merely topical: both papers study how auxiliary high-dimensional Ising tasks can be exploited to improve estimation for a target Ising model; both adopt nodewise $\\ell_1$-regularized logistic regression as the main estimation tool; both use auxiliary tasks to recover a larger transferable support set or shared structural information before refining estimation on the target task; and both establish theory of essentially the same flavor, including support recovery and improved sample-complexity guarantees under standard high-dimensional Ising assumptions.

The transfer-Ising paper does add practically useful components such as source detection and a refined second-stage correction, so it is not literally identical. However, from the perspectives of problem formulation, methodological backbone, and theoretical flavor, the contribution appears to lie more in adapting and refining existing ideas in the Ising-model setting than in introducing a clearly new conceptual advance.

[1] Tian, Y. and Y. Feng (2023). Transfer learning under high-dimensional generalized linear models. *Journal of the American Statistical Association* 118 (544), 2684–2697.

[2] Ravikumar, P., M. J. Wainwright, and J. D. Lafferty (2010). High-dimensional Ising model selection using $\\ell_1$-regularized logistic regression. *The Annals of Statistics* 38 (3), 1287–1319.

[3] Xie, H. and J. Honorio (2024). Meta learning for support recovery of high-dimensional Ising models. *Transactions on Machine Learning Research*.

---

> ### Author Rebuttal · Authors · 2026-03-31
>
> We thank the reviewer for the careful evaluation and address the main concerns below.
> # Q1: Methodological differences from Tian & Feng (2023)
> We agree that the distinction from Tian & Feng (2023) should be stated more clearly. Although both papers share the broad “pool then correct” blueprint, our method differs in three Ising-specific ways.
> - **Dual regularization for exact support recovery.** Tian & Feng use a purely $\ell_1$-based correction step. Our Step 2 instead uses a dual penalty ($\ell_1$ on the correction and SCAD on the updated coefficients), which allows us to prove exact support recovery of the final transferred estimator without an irrepresentable condition. This is not a routine replacement of $\ell_1$ by SCAD: the proof must work with a nonconvex objective and a computable stationary local solution, and uses the SCAD flat region to eliminate shrinkage bias on the true support.
> - **Pseudolikelihood-based source detection in the Ising setting.** Our screening rule is based on held-out target pseudolikelihood rather than a standard GLM loss, because the joint Ising likelihood contains an intractable partition function. The corresponding consistency proof exploits the specific structure of the Ising conditional model and differs from the argument in Tian & Feng.
> - **Nodewise decomposition and graph-level symmetrization.** In the Ising model, estimation proceeds through $p$ nodewise regressions followed by graph symmetrization. This creates a graph-recovery problem rather than a single-regression parameter-estimation problem, and requires uniform control across all nodes to obtain graph-level guarantees.
> # Q2: Relationship to Xie & Honorio (2024)
> We agree this paper is highly relevant and should be discussed in detail. We believe the two works address *different transfer regimes*.
>
> |**Aspect**|**Xie & Honorio**|**Ours**|
> |:-|:-|:-|
> |Learning setting|Meta-learning over many related Ising tasks from a shared generative family|Transfer learning from heterogeneous auxiliary sources of unknown relevance|
> |Task assumption|Every task support is contained in a common latent support union; all tasks belong to the same meta-family|A subset of sources is assumed close to the target; other sources may be harmful|
> |Auxiliary-task goal|Recover a common support union from all tasks, then restrict the novel-task estimator to that union|Screen sources, construct a pooled initializer from selected sources, then perform a target-only correction|
> |Unknown source relevance|Not part of the formulation; all auxiliary tasks are assumed relevant samples from the meta-family|Central part of the formulation; screening is needed to avoid negative transfer|
> |Main theory|Sample-complexity guarantees for support-union recovery and restricted novel-task neighborhood recovery|Transfer-type estimation rate, exact support/sign recovery, and source-detection consistency|
> |Regularization|$\ell_1$-regularized logistic regression|$\ell_1$ correction + SCAD on the updated coefficients|
>
> Although the two papers operate in a related high-dimensional Ising setting, they address different problem regimes and objectives. We therefore view the two papers as complementary rather than redundant. We will revise the related work section including this.
> # Q3: Robustness beyond the original simulation regime
> We agree this is important, and ran three additional sweeps over $\sigma$, $n_0$, and $n_s$ (all at $p=200$ with 3 informative and 9 non-informative sources; relative Frobenius error over 50 replications).
>
> **(a) Source-target discrepancy** ($n_0=160$; $n_s=300$):
>
> |$\sigma$|Oracle|Trans|Naive|Pooled|
> |:-:|:-:|:-:|:-:|:-:|
> |0.01|0.433|0.433|0.891|0.666|
> |0.10|0.441|0.442|0.890|0.668|
> |0.20|0.471|0.471|0.891|0.687|
> |0.50|0.754|0.829|0.891|0.849|
> |1.00|0.979|0.891|0.891|0.952|
>
> This gives a clear ability boundary. Transfer is
> strongly helpful up to about $\sigma\approx 0.2$. At
> $\sigma=0.5$, the heterogeneity begins to offset the benefit of
> variance reduction and screening errors become visible. By
> $\sigma=1.0$, transfer is no longer beneficial even for the nominally
> informative sources, and Trans-Ising essentially rejects them and
> reverts to the target-only baseline.
>
> **(b) Target sample size** ($n_s=300$; $\sigma=0.1$):
>
> |$n_0$|Oracle|Trans|Naive|Pooled|
> |:-:|:-:|:-:|:-:|:-:|
> |80 |0.461|0.461|0.966|0.758|
> |120|0.449|0.448|0.929|0.709|
> |160|0.441|0.441|0.889|0.673|
> |300|0.427|0.427|0.789|0.572|
>
> The gain from transfer is largest when the target sample is most limited, and narrows as $n_0$ increases, consistent with Theorem 1.
>
> **(c) Per-source auxiliary sample size** ($n_0=160$; $\sigma=0.1$):
>
> |$n_s$|Oracle|Trans|Naive|Pooled|
> |:-:|:-:|:-:|:-:|:-:|
> |100|0.577|0.577|0.891|0.724|
> |200|0.487|0.486|0.892|0.690|
> |300|0.441|0.441|0.891|0.672|
> |500|0.397|0.397|0.890|0.653|
>
> As $n_s$ increases, both Oracle and Trans-Ising improve monotonically, consistent with Theorem 1, with diminishing returns once heterogeneity becomes the limiting factor.

---

> > ### Author Rebuttal · Reviewer_vyew · 2026-04-04
> >
> > Thanks to the authors for their thoughtful response to my questions. However, I remain unconvinced about the originality of the work. In my view, transfer-Ising seems largely to combine the ideas of Tian and Feng (2023) and Ravikumar et al. (2010), particularly given that a very similar heterogeneous case has already been addressed by Xie and Honorio (2024). As such, I continue to have reservations about the marginal contribution of this paper.

---

> > > ### Author Response · Authors · 2026-04-04
> > >
> > > We sincerely thank the reviewer for the continued engagement. We would like to respectfully clarify one factual point: Xie & Honorio (2024) assume all auxiliary tasks are drawn from a shared generative family with a common support union (Definition 3.1 in their paper), and do not address unknown source relevance or negative transfer. In contrast, our formulation explicitly allows harmful sources and provides a provably consistent screening mechanism (Theorem 3) to exclude them. We believe this is a fundamental difference in problem formulation rather than a minor variation.
> > >
> > > Regarding the characterization as combining Tian & Feng (2023) and Ravikumar et al. (2010): we respectfully note that this characterization could similarly apply to many contributions in high-dimensional transfer learning, where methodological advances often build on established estimation tools and transfer principles. We believe the appropriate criterion is whether the resulting theory addresses problems that neither prior work alone can solve. Our Theorem 2 (selection consistency without the irrepresentable condition) and Theorem 3 (source-detection consistency under Ising pseudolikelihood) are results that do not follow from specializing either prior framework, as detailed in our initial rebuttal.
> > >
> > > We also hope that the expanded sensitivity analysis provided in our initial rebuttal adequately addresses the concern regarding simulation robustness across different regimes of target sample size, auxiliary sample size, and source-target discrepancy.
> > >
> > > We fully respect the reviewer's assessment and hope these clarifications are helpful to the discussion.

---

### Official Review · Reviewer_Qcx8 · 2026-03-13

**Soundness:** 3
**Presentation:** 3
**Significance:** 3
**Originality:** 3
**Overall Recommendation:** 5
**Confidence:** 5

**Summary:**

This manuscript focuses on a transfer learning task in high-dimensional Ising models, where the parameters are assumed to be sparse and the parameter is similar to the one from the "informative source dataset". Similar to a transfer learning work in genearlized linear models (Tian and Feng 2023), authors propose a two step procedure comprised of "transferring step" in eq. 8 and a "debiasing step" in eq. 9, where SCAD penalty (Fan and Li 2001) in addition to $\ell_1$ penality is introduced.
Authors introduce two version of the algorithms; first the Oracle trans-Ising algorithm with a known informative source set (Algorithm 1) and a trans-Ising algorithm where informative soucres are selected based on held-out target pseudolikelihood. (Algorithm 2). A key contribution is a theoretical analysis given in Section 4. In Theorem 1, authors provide a rate analysis in terms of $N = n_0 + \sum_{s\in \mathcal{A}}n_s$ which improves a target-only rate based on $n_0$. Theorem 2 describes a selection consistency under suitable regularity conditions appropriate for SCAD penalty, and Theorem 3 describes a consistency of source detection.  Authors carry out an extensive empirical study based on synthetic datasets as well as three real data with mutational data, online transaction anaylsis, and movie data analysis.


* Tian, Y., & Feng, Y. (2023). Transfer learning under high-dimensional generalized linear models. Journal of the American Statistical Association, 118(544), 2684-2697.

* Fan, J., & Li, R. (2001). Variable selection via nonconcave penalized likelihood and its oracle properties. Journal of the American statistical Association, 96(456), 1348-1360.

**Compliance With Llm Reviewing Policy:**

Affirmed.

**Final Justification:**

The author's rebuttal well addressed my main concerns on computational difficulty associated with SCAD penalty, which reinforced my prior assessment, and I am updating my confidence from 4 to 5 accordingly.

**Key Questions For Authors:**

Below comments are mostly minor.

* In line 64, authors argue the selection consistency is established under a *weaker* beta-min condition; could you elaborate on the "weaker" argument?

* Having consistent notation for indices would greatly help readability; e.g. in eq. 4 $(i,j)$ is used to represent edges but $i$ in eq. 8 is used for indexing samples and $(j,k)$ is used for edges.

* I found eq. 6's notation with 0/1 response setting and Remark 1 a bit confusing due to incompatible loss function form between eq. 8. I believe revising eq. 6. using an expression based on $f(z)=\log(1+e^{-z})$ similar to eq. 8 using -1/1 response setting, as well as $n_0$ instead of $n$ would improve readablity.

* I personally felt the symmetrization procedure described in line 5 of Algorithm 1 somewhat ad-hoc; if a similar workaround has been proposed in the literature, please mention it.

* in line 367, authors say "While the strategy of cross-validation based screening shares the spirit of Tian & Feng (2023), establishing its consistency in Ising models presents unique theoretical challenges. Unlike GLMs where samples are typically assumed independent, the Ising model involves complex dependencies among variables. ". Could you elaborate this? It appears that proof is actually more simpler than Tian & Feng (2023)'s case since all variables are -1 or 1, and the samples are also independent across $i=1,...n$.

* In the empirical result presented in the article, the performance between oracle trans Ising and trans Ising is identical from Figure 1 and 2, which i found surprising. This makes me think about whether simulation design isn't too optimistic and question about what circumstances would cause the Trans-Ising perform poorer than the oracle settings.

* In remark 7, the authors emphasize SCAD penalty enables weaker assumption on design matrix than lasso but the reason could be better clarified. Reference to the existing literature including the relevant discussion  or ideally a concrete example similar to Sec 3.4.2 of Yang et al. (2016) would be extremely valuable.

Yang, Y., Wainwright, M. J., & Jordan, M. I. (2016). ON THE COMPUTATIONAL COMPLEXITY OF HIGH-DIMENSIONAL BAYESIAN VARIABLE SELECTION. The Annals of Statistics, 2497-2532.

**Limitations:**

There is no potential negative societal impact of their work.

**Strengths And Weaknesses:**

The manuscript is well written and presents a valuable contribution to the growing field of transfer learning in high-dimensional statistics. The proposed algorithms, albeit conceptually similar to GLM transfer learning setting of (Tian and Feng 2023), is reasonable and the empirical study provides clear benefits over a pooled model. One of the key contributions is an additional scad penalty described in eq. 9, which allows a theory without an irrepresentable condition that is both necessary and sufficient under an $\ell_1$ penalty to establish selection consistency.

While the manuscript is overall strong, its limitations could be better articulated. The authors highly emphasize the benefit of this SCAD penalty but put much less emphasis on the computational tradeoff; e.g. assumption 6 is deferred to the appendix. The approximate optimization strategy based on local linear approximation appears to be a feasible option, but a short justification or ideally a comparison with the penalized linear unbiased selection (PLUS) algorithm of Zhang (2010) would provide a valuable insight to readers.

* ZHANG, B. C. H. (2010). NEARLY UNBIASED VARIABLE SELECTION UNDER MINIMAX CONCAVE PENALTY. The Annals of Statistics, 38(2), 894-942.

---

> ### Author Rebuttal · Authors · 2026-03-31
>
> We sincerely appreciate the reviewer’s supportive assessment and thoughtful, technically precise suggestions.
> We address each point below.
> # Q1: About the “weaker beta-min condition”
> Thank you for pointing this out. The intended meaning of “weaker” was that the *overall structural conditions* for selection consistency are weaker. In standard Lasso-based neighborhood selection, exact support recovery typically requires both a beta-min condition and an irrepresentable (mutual incoherence) condition on the Hessian. By contrast, our Theorem 2 is established under target restricted strong convexity (RSC), a standard beta-min separation, and a computable stationary local solution, without imposing an irrepresentable condition.
>
> So the precise “weaker” argument is: the beta-min condition is still standard, but the accompanying design-side assumptions are weaker because the irrepresentable condition is not required. We will revise the wording accordingly to: *selection consistency under weaker design assumptions, together with a standard beta-min condition*.
>
> # Q2: Consistent notation for indices
> We thank the reviewer for catching this.
> We will unify the index convention throughout the paper ($i$ for samples, $(j,k)$ for nodes/edges) and conduct a systematic proofreading pass to ensure no instances are missed.
> # Q3: Eq. 6 / Remark 1 notation mismatch
> We agree that mixing the encodings in the main exposition is confusing.
> In the revision we will rewrite the nodewise loss directly in the signed-design form used by the estimator and the proofs, so that Eq. 6 and Eq. 8 match exactly
> and move the $0/1$ logistic form to the appendix.
> # Q4: Symmetrization procedure
> We agree that the rationale should be stated more explicitly. The AND
> symmetrization rule is standard in neighborhood-selection-based graph
> recovery.
> We chose AND, rather than OR, because it is the conservative rule: once
> we have uniform nodewise sign consistency, AND preserves false-positive
> control and yields exact graph recovery at the graph level.
> We will add these justifications in Section 3.
> # Q5: “Unique theoretical challenges” relative to Tian & Feng (2023)
> The reviewer is right that the samples are i.i.d. across $i$ in our Ising setting. What we intended to highlight was not sample-level dependence, but the dependence structure *within* each observation. For a fixed sample $x_i \in$ {-1,1}$^p$, the coordinates used in the nodewise regression for node $j$ are jointly distributed under the Ising model, so the analysis is carried out through the conditional / pseudolikelihood representation.
>
> We will revise the sentence accordingly. The key point is that our source-detection theorem controls held-out target pseudolikelihood differences under the bounded signed {-1,1} design of the Ising nodewise model, whereas Tian & Feng (2023) work under explicit sub-Gaussian predictor assumptions for GLMs. Thus, the high-level screening logic is similar, but the concentration arguments are adapted to a different structural setting.
>
> # Q6: Oracle and Trans-Ising appearing identical
> Thank you for this helpful comment. To test a harder regime, we extended the perturbation sweep to $\sigma\in$ {0.01,0.1,0.2,0.5,1.0} at $p=200$, $n_0=160$, $n_s=300$, with 3 informative and 9 non-informative sources.
> The relative Frobenius errors are:
>
> |$\sigma$|Oracle|Trans|Naive|Pooled|
> |:-:|:-:|:-:|:-:|:-:|
> |0.01|0.433|0.433|0.891|0.666|
> |0.10|0.441|0.442|0.890|0.668|
> |0.20|0.471|0.471|0.891|0.687|
> |0.50|0.754|0.829|0.891|0.849|
> |1.00|0.979|0.891|0.891|0.952|
>
> This clarifies the pattern. For relatively small discrepancy ($\sigma\le 0.2$), screening remains stable and Trans-Ising nearly coincides with the oracle. At $\sigma=0.5$, screening errors begin to matter and Trans-Ising becomes noticeably worse than the Oracle. At $\sigma=1.0$, even the nominally informative sources are too heterogeneous to help, so the oracle deteriorates, while Trans-Ising screens them out and reverts to the target-only baseline. We will add this expanded analysis to the paper.
>
> # Q7: Why SCAD relaxes the design requirement relative to Lasso
> We agree that this point deserves a sharper explanation and a concrete example. Relative to Lasso, the key difference is that our SCAD-based result does not require an irrepresentable condition for support recovery.
> In transfer learning, the pooled initialization can make Lasso-based recovery more sensitive to off-support correlations, and hence to violations of the irrepresentable condition. By contrast, for SCAD the penalty derivative becomes zero once a coefficient enters the flat region, so the regularization bias on sufficiently large active coordinates disappears. This is why folded-concave penalties can achieve support recovery under RSC-type conditions without an irrepresentable assumption.
> In the revision we will add a short example showing a regime where the transfer offset can violate the Lasso irrepresentable condition, while the target RSC condition still holds.

---

> > ### Author Rebuttal · Reviewer_Qcx8 · 2026-04-02
> >
> > I acknowledge the author's rebuttal, the additional simulation analysis clarifying the difference between Oracle and trans-Ising and its explanation is helpful.
> >
> > > While the manuscript is overall strong, its limitations could be better articulated. The authors highly emphasize the benefit of this SCAD penalty but put much less emphasis on the computational tradeoff; e.g. assumption 6 is deferred to the appendix. The approximate optimization strategy based on local linear approximation appears to be a feasible option, but a short justification or ideally a comparison with the penalized linear unbiased selection (PLUS) algorithm of Zhang (2010) would provide a valuable insight to readers. ZHANG, B. C. H. (2010). NEARLY UNBIASED VARIABLE SELECTION UNDER MINIMAX CONCAVE PENALTY. The Annals of Statistics, 38(2), 894-942.
> >
> > I would appreciate it if authors could also discuss my previous point in depth (quoted above). Specifically, could you compare the local linear approximation scheme proposed in this paper and the penalized linear unbiased selection (PLUS) algorithm of Zhang (2010) ?

---

> > > ### Author Response · Authors · 2026-04-03
> > >
> > > We are sincerely grateful to the reviewer for the thoughtful follow-up
> > > and for bringing the PLUS algorithm of Zhang (2010) to our attention.
> > > The suggestion to compare our LLA-based optimization with PLUS will meaningfully strengthen our paper.
> > > # Comparison of LLA and PLUS
> > > Both LLA (Zou & Li, 2008) and PLUS (Zhang, 2010) address
> > > nonconvex penalized regression and aim to mitigate the shrinkage bias
> > > inherent in $\ell_1$ regularization, but they differ in mechanism,
> > > scope, and the guarantees:
> > >
> > > - **LLA** is a majorization-minimization scheme applicable to general
> > > loss functions combined with a broad class of folded-concave penalties.
> > > At each iteration, it linearizes the concave penalty at the current estimate, producing adaptive weights
> > > $\omega_k^{(t)} = P'_\lambda(|\vartheta_k^{(t)}|)$, and solves a
> > > reweighted $\ell_1$ (convex) surrogate. Convergence is to a stationary
> > > point satisfying approximate KKT conditions, which is precisely what
> > > our Assumption 6 formalizes.
> > >
> > > - **PLUS** extends the LARS path algorithm to quadratic spline penalties
> > > (including both SCAD and MCP) by tracing a continuous,
> > > piecewise-linear solution path from zero coefficients to the
> > > unpenalized solution. Under the sparse Riesz condition (SRC), PLUS
> > > identifies a distinguished local minimizer on the main branch of the
> > > solution path that provably possesses oracle properties (sign
> > > consistency and minimax-rate convergence). This is a *local*
> > > minimizer with strong statistical guarantees, not necessarily the
> > > global minimizer of the nonconvex objective.
> > >
> > > An important practical distinction is that PLUS relies on
> > > the piecewise-linear solution path structure inherited from LARS, which
> > > **requires a least-squares loss function**. In our setting, Step 2
> > > minimizes a logistic pseudolikelihood loss, for which the LARS-type path is not
> > > piecewise-linear and the PLUS construction does not directly apply. This
> > > is the primary reason we adopted LLA, which accommodates general
> > > differentiable loss functions.
> > >
> > > Beyond the loss-function constraint, two additional considerations favor LLA in our framework:
> > >
> > > 1. Our Step 2 objective involves a *dual* penalty structure,
> > > rather than a single-penalty formulation. LLA naturally handles this composite structure: at each iteration, the SCAD component is
> > > linearized into adaptive weights while the $\ell_1$ term on $\delta$
> > > remains unchanged, yielding a convex surrogate solvable via proximal
> > > gradient methods or coordinate descent (e.g., `glmnet`).
> > > 2. From a theoretical standpoint, our Theorems 1 and 2 require only
> > > that the solver returns an approximate stationary point in a local
> > > neighborhood of the oracle solution (Assumption 6)—a condition
> > > that LLA is designed to satisfy. The stronger path-based guarantees
> > > of PLUS, while theoretically appealing, are not strictly required
> > > for our current results, though they would be a welcome
> > > strengthening in future extensions of the framework.
> > >
> > > That said, PLUS offers a valuable conceptual benchmark. Under the sparse
> > > Riesz condition, the PLUS solution on the main branch achieves oracle
> > > properties without the irrepresentable condition—paralleling our own
> > > irrepresentable-free guarantee via SCAD's flat-penalty-derivative
> > > region. We also note that coordinate descent algorithms for nonconvex
> > > penalties (Breheny & Huang, 2011) provide another efficient alternative
> > > applicable to generalized linear models, and may offer computational
> > > advantages for large-scale problems. In the revision, we will include an
> > > explicit discussion of these algorithmic alternatives, clarifying when each is applicable and the
> > > tradeoffs involved.
> > > # Assumption 6 and computational transparency
> > > We appreciate the reviewer's observation that Assumption 6 is
> > > deferred to the appendix, which may inadvertently downplay the
> > > computational considerations underlying our framework. Assumption 6
> > > requires that the optimization routine returns a solution satisfying
> > > three conditions: (i) approximate stationarity (KKT residual bounded by
> > > $\varepsilon_{n,j}$), (ii) local minimality within an $\ell_1$-ball of
> > > radius $r_\delta$ around the oracle value, and (iii) membership in the
> > > RSC neighborhood specified by Assumption 4. These
> > > conditions formalize the gap between the statistical theory (which
> > > analyzes a local solution) and the computational practice (which depends
> > > on the solver and initialization quality).
> > >
> > > We agree that this deserves more prominent treatment. In the revision,
> > > we will: (1) state Assumption 6 in the main text alongside the
> > > description of Step 2 in Algorithm 1, with a clear explanation of what
> > > it requires and why it is needed; (2) add a remark noting that the
> > > pooled estimator from Step 1 provides a natural warm start that, when
> > > close to the target oracle value, helps LLA converge to a local solution
> > > satisfying Assumption 6; and (3) include the comparison of LLA, PLUS,
> > > and coordinate descent so readers can assess both
> > > the statistical and computational aspects of our framework.
> > >
> > > We thank the reviewer again for these valuable suggestions.

---

### Decision · Program_Chairs · 2026-04-30

**Decision:**

Accept (regular)

**Comment:**

This paper studies transfer learning in high-dimensional Ising models, aiming to estimate a sparse target network using limited target data and multiple auxiliary datasets of unknown relevance. The paper proposes Trans-Ising, a two-stage framework: a transfer step that pools selected auxiliary data via regularized nodewise logistic regression, and a debiasing step combining $\ell_1$ and SCAD penalties to reduce shrinkage bias and improve support recovery. A data-driven source selection method based on held-out pseudolikelihood mitigates negative transfer. The paper provides theoretical guarantees for estimation error, support recovery, and source selection consistency, and demonstrates strong empirical performance on synthetic and real datasets.

The main point of discussion in this paper is its distinction from Xie & Honorio (2024). The key difference from Xie & Honorio (2024) lies in the problem setting, while the distinction from Tian and Feng (2023) is the choice of regularization ($\ell_1$ vs. SCAD). The authors further establish selection consistency in Theorem 2, and the experimental results provide strong support for this theoretical finding. Overall, the contribution of this paper is solid and meaningful. Therefore, I recommend accepting this paper.